# Learning Time-Varying Coverage Functions

**Nan Du[†], Yingyu Liang[‡], Maria-Florina Balcan[◇], Le Song[†]**
[†]College of Computing, Georgia Institute of Technology
[‡]Department of Computer Science, Princeton University
[◇]School of Computer Science, Carnegie Mellon University
dunan@gatech.edu,yingyul@cs.princeton.edu
ninamf@cs.cmu.edu,lsong@cc.gatech.edu

## Abstract

Coverage functions are an important class of discrete functions that capture the law of diminishing returns arising naturally from applications in social network analysis, machine learning, and algorithmic game theory. In this paper, we propose a new problem of learning time-varying coverage functions, and develop a novel parametrization of these functions using random features. Based on the connection between time-varying coverage functions and counting processes, we also propose an efficient parameter learning algorithm based on likelihood maximization, and provide a sample complexity analysis. We applied our algorithm to the influence function estimation problem in information diffusion in social networks, and show that with few assumptions about the diffusion processes, our algorithm is able to estimate influence significantly more accurately than existing approaches on both synthetic and real world data.

## 1   Introduction

Coverage functions are a special class of the more general submodular functions which play important role in combinatorial optimization with many interesting applications in social network analysis [1], machine learning [2], economics and algorithmic game theory [3], etc. A particularly important example of coverage functions in practice is the influence function of users in information diffusion modeling [1] — news spreads across social networks by word-of-mouth and a set of influential sources can collectively trigger a large number of follow-ups. Another example of coverage functions is the valuation functions of customers in economics and game theory [3] — customers are thought to have certain requirements and the items being bundled and offered fulfill certain subsets of these demands.

Theoretically, it is usually assumed that users' influence or customers' valuation are known in advance as an oracle. In practice, however, these functions must be learned. For example, given past traces of information spreading in social networks, a social platform host would like to estimate how many follow-ups a set of users can trigger. Or, given past data of customer reactions to different bundles, a retailer would like to estimate how likely customer would respond to new packages of goods. Learning such combinatorial functions has attracted many recent research efforts from both theoretical and practical sides (*e.g.*, [4, 5, 6, 7, 8]), many of which show that coverage functions can be learned from just polynomial number of samples.

However, the prior work has widely ignored an important dynamic aspect of the coverage functions. For instance, information spreading is a dynamic process in social networks, and the number of follow-ups of a fixed set of sources can increase as observation time increases. A bundle of items or features offered to customers may trigger a sequence of customer actions over time. These real world problems inspire and motivate us to consider a novel *time-varying coverage function*, $f(\mathcal{S}, t)$, which is a coverage function of the set $\mathcal{S}$ when we fix a time $t$, and a continuous monotonic function of time $t$ when we fix a set $\mathcal{S}$. While learning time-varying combinatorial structures has been ex-

plored in graphical model setting (*e.g.*, [9, 10]), as far as we are aware of, learning of time-varying coverage function has not been addressed in the literature. Furthermore, we are interested in estimating the entire function of $t$, rather than just treating the time $t$ as a discrete index and learning the function value at a small number of discrete points. From this perspective, our formulation is the generalization of the most recent work [8] with even less assumptions about the data used to learn the model.

Generally, we assume that the historical data are provided in pairs of a set and a collection of timestamps when caused events by the set occur. Hence, such a collection of temporal events associated with a particular set $\mathcal{S}_i$ can be modeled principally by a counting process $N_i(t), t \geqslant 0$ which is a stochastic process with values that are positive, integer, and increasing along time [11]. For instance, in the information diffusion setting of online social networks, given a set of earlier adopters of some new product, $N_i(t)$ models the time sequence of all triggered events of the followers, where each jump in the process records the timing $t_{ij}$ of an action. In the economics and game theory setting, the counting process $N_i(t)$ records the number of actions a customer has taken over time given a particular bundled offer. This essentially raises an interesting question of how to estimate the time-varying coverage function from the angle of counting processes. We thus propose a novel formulation which builds a connection between the two by modeling the cumulative intensity function of a counting process as a time-varying coverage function. The key idea is to parametrize the intensity function as a weighted combination of random kernel functions. We then develop an efficient learning algorithm TCOVERAGELEARNER to estimate the parameters of the function using maximum likelihood approach. We show that our algorithm can provably learn the time-varying coverage function using only polynomial number of samples. Finally, we validate TCOVERAGELEARNER on both influence estimation and maximization problems by using cascade data from information diffusion. We show that our method performs significantly better than alternatives with little prior knowledge about the dynamics of the actual underlying diffusion processes.

## 2   Time-Varying Coverage Function

We will first give a formal definition of the time-varying coverage function, and then explain its additional properties in details.

**Definition**. Let $\mathcal{U}$ be a (potentially uncountable) domain. We endow $\mathcal{U}$ with some $\sigma$-algebra $\mathscr{A}$ and denote a probability distribution on $\mathcal{U}$ by $\mathbb{P}$. A coverage function is a combinatorial function over a finite set $\mathcal{V}$ of items, defined as

$$f(\mathcal{S}) := Z \cdot \mathbb{P}\left(\bigcup_{s \in \mathcal{S}} \mathcal{U}_s\right), \quad \text{for all } \mathcal{S} \in 2^{\mathcal{V}}, \tag{1}$$

where $\mathcal{U}_s \subset \mathcal{U}$ is the subset of domain $\mathcal{U}$ covered by item $s \in \mathcal{V}$, and $Z$ is the additional normalization constant. For time-varying coverage functions, we let the size of the subset $\mathcal{U}_s$ to grow monotonically over time, that is

$$\mathcal{U}_s(t) \subseteq \mathcal{U}_s(\tau), \quad \text{for all } t \leqslant \tau \text{ and } s \in \mathcal{V}, \tag{2}$$

which results in a combinatorial temporal function

$$f(\mathcal{S}, t) = Z \cdot \mathbb{P}\left(\bigcup_{s \in \mathcal{S}} \mathcal{U}_s(t)\right), \quad \text{for all } \mathcal{S} \in 2^{\mathcal{V}}. \tag{3}$$

In this paper, we assume that $f(\mathcal{S}, t)$ is smooth and continuous, and its first order derivative with respect to time, $f'(\mathcal{S}, t)$, is also smooth and continuous.

**Representation**. We now show that a time-varying coverage function, $f(\mathcal{S}, t)$, can be represented as an expectation over random functions based on multidimensional step basis functions. Since $\mathcal{U}_s(t)$ is varying over time, we can associate each $u \in \mathcal{U}$ with a $|\mathcal{V}|$-dimensional vector $\boldsymbol{\tau}_u$ of change points. In particular, the $s$-th coordinate of $\boldsymbol{\tau}_u$ records the time that source node $s$ covers $u$. Let $\boldsymbol{\tau}$ to be a random variable obtained by sampling $u$ according to $\mathbb{P}$ and setting $\boldsymbol{\tau} = \boldsymbol{\tau}_u$. Note that given all $\boldsymbol{\tau}_u$ we can compute $f(\mathcal{S}, t)$; now we claim that the distribution of $\boldsymbol{\tau}$ is sufficient.

We first introduce some notations. Based on $\boldsymbol{\tau}_u$ we define a $|\mathcal{V}|$-dimensional step function $\boldsymbol{r}_u(t) :$ $\mathbb{R}_+ \mapsto \{0,1\}^{|\mathcal{V}|}$, where the $s$-th dimension of $\boldsymbol{r}_u(t)$ is 1 if $u$ is covered by the set $\mathcal{U}_s(t)$ at time $t$, and 0 otherwise. To emphasize the dependence of the function $\boldsymbol{r}_u(t)$ on $\boldsymbol{\tau}_u$, we will also write $\boldsymbol{r}_u(t)$ as $\boldsymbol{r}_u(t|\boldsymbol{\tau}_u)$. We denote the indicator vector of a set $\mathcal{S}$ by $\boldsymbol{\chi}_{\mathcal{S}} \in \{0,1\}^{|\mathcal{V}|}$ where the $s$-th dimension of $\boldsymbol{\chi}_{\mathcal{S}}$ is 1 if $s \in \mathcal{S}$, and 0 otherwise. Then $u \in \mathcal{U}$ is covered by $\bigcup_{s \in \mathcal{S}} \mathcal{U}_s(t)$ at time $t$ if $\boldsymbol{\chi}_{\mathcal{S}}^\top \boldsymbol{r}_u(t) \geqslant 1$.

**Lemma 1.** *There exists a distribution $\mathbb{Q}(\boldsymbol{\tau})$ over the vector of change points $\boldsymbol{\tau}$, such that the time-varying coverage function can be represented as*

$$f(\mathcal{S},t) = Z \cdot \mathbb{E}_{\boldsymbol{\tau} \sim \mathbb{Q}(\boldsymbol{\tau})} \left[ \phi(\boldsymbol{\chi}_\mathcal{S}^\top \boldsymbol{r}(t|\boldsymbol{\tau})) \right] \tag{4}$$

*where $\phi(x) := \min\{x, 1\}$, and $\boldsymbol{r}(t|\boldsymbol{\tau})$ is a multidimensional step function parameterized by $\boldsymbol{\tau}$.*

*Proof.* Let $\mathcal{U}_\mathcal{S} := \bigcup_{s \in \mathcal{S}} \mathcal{U}_s(t)$. By definition (3), we have the following integral representation

$$f(\mathcal{S},t) = Z \cdot \int_{\mathcal{U}} \mathbb{I}\left\{u \in \mathcal{U}_\mathcal{S}\right\} d\mathbb{P}(u) = Z \cdot \int_{\mathcal{U}} \phi(\boldsymbol{\chi}_\mathcal{S}^\top \boldsymbol{r}_u(t)) \, d\mathbb{P}(u) = Z \cdot \mathbb{E}_{u \sim \mathbb{P}(u)} \left[ \phi(\boldsymbol{\chi}_\mathcal{S}^\top \boldsymbol{r}_u(t)) \right].$$

We can define the set of $u$ having the same $\boldsymbol{\tau}$ as $\mathcal{U}_{\boldsymbol{\tau}} := \{u \in \mathcal{U} \mid \tau_u = \boldsymbol{\tau}\}$ and define a distribution over $\boldsymbol{\tau}$ as $d\mathbb{Q}(\boldsymbol{\tau}) := \int_{\mathcal{U}_{\boldsymbol{\tau}}} d\mathbb{P}(u)$. Then the integral representation of $f(\mathcal{S},t)$ can be rewritten as

$$Z \cdot \mathbb{E}_{u \sim \mathbb{P}(u)} \left[ \phi(\boldsymbol{\chi}_\mathcal{S}^\top \boldsymbol{r}_u(t)) \right] = Z \cdot \mathbb{E}_{\boldsymbol{\tau} \sim \mathbb{Q}(\boldsymbol{\tau})} \left[ \phi(\boldsymbol{\chi}_\mathcal{S}^\top \boldsymbol{r}(t|\boldsymbol{\tau})) \right],$$

which proves the lemma. $\qquad\square$

## 3 Model for Observations

In general, we assume that the input data are provided in the form of pairs, $(\mathcal{S}_i, N_i(t))$, where $\mathcal{S}_i$ is a set, and $N_i(t)$ is a counting process in which each jump of $N_i(t)$ records the timing of an event. We first give a brief overview of a counting process [11] and then motivate our model in details.

**Counting Process**. Formally, a counting process $\{N(t), t \geqslant 0\}$ is any nonnegative, integer-valued stochastic process such that $N(t') \leqslant N(t)$ whenever $t' \leqslant t$ and $N(0) = 0$. The most common use of a counting process is to count the number of occurrences of temporal events happening along time, so the index set is usually taken to be the nonnegative real numbers $\mathbb{R}_+$. A counting process is a submartingale: $\mathbb{E}[N(t) \mid \mathcal{H}_{t'}] \geqslant N(t')$ for all $t > t'$ where $\mathcal{H}_{t'}$ denotes the history up to time $t'$. By Doob-Meyer theorem [11], $N(t)$ has the unique decomposition:

$$N(t) = \Lambda(t) + M(t) \tag{5}$$

where $\Lambda(t)$ is a nondecreasing predictable process called the compensator (or cumulative intensity), and $M(t)$ is a mean zero martingale. Since $\mathbb{E}[dM(t) \mid \mathcal{H}_{t-}] = 0$, where $dM(t)$ is the increment of $M(t)$ over a small time interval $[t, t + dt)$, and $\mathcal{H}_{t-}$ is the history until just before time $t$,

$$\mathbb{E}[dN(t) \mid \mathcal{H}_{t-}] = d\Lambda(t) := a(t) \, dt \tag{6}$$

where $a(t)$ is called the intensity of a counting process.

**Model formulation**. We assume that the cumulative intensity of the counting process is modeled by a time-varying coverage function, *i.e.*, the observation pair $(\mathcal{S}_i, N_i(t))$ is generated by

$$N_i(t) = f(\mathcal{S}_i, t) + M_i(t) \tag{7}$$

in the time window $[0, T]$ for some $T > 0$, and $df(\mathcal{S}, t) = a(\mathcal{S}, t)dt$. In other words, the time-varying coverage function controls the propensity of occurring events over time. Specifically, for a fixed set $\mathcal{S}_i$, as time $t$ increases, the cumulative number of events observed grows accordingly for that $f(\mathcal{S}_i, t)$ is a continuous monotonic function over time; for a given time $t$, as the set $\mathcal{S}_i$ changes to another set $\mathcal{S}_j$, the amount of coverage over domain $\mathcal{U}$ may change and hence can result in a different cumulative intensity. This abstract model can be mapped to real world applications. In the information diffusion context, for a fixed set of sources $\mathcal{S}_i$, as time $t$ increases, the number of influenced nodes in the social network tends to increase; for a given time $t$, if we change the sources to $\mathcal{S}_j$, the number of influenced nodes may be different depending on how influential the sources are. In the economics and game theory context, for a fixed bundle of offers $\mathcal{S}_i$, as time $t$ increases, it is more likely that the merchant will observe the customers' actions in response to the offers; even at the same time $t$, different bundles of offers, $\mathcal{S}_i$ and $\mathcal{S}_j$, may have very different ability to drive the customers' actions.

Compared to a regression model $y_i = g(\mathcal{S}_i) + \epsilon_i$ with *i.i.d.* input data $(\mathcal{S}_i, y_i)$, our model outputs a special random function over time, that is, a counting process $N_i(t)$ with the noise being a zero mean martingale $M_i(t)$. In contrast to functional regression models, our model exploits much more interesting structures of the problem. For instance, the random function representation in the last section can be used to parametrize the model. Such special structure of the counting process allows us to estimate the parameter of our model using maximum likelihood approach efficiently, and the martingale noise enables us to use exponential concentration inequality in analyzing our algorithm.

## 4 Parametrization

Based on the following two mild assumptions, we will show how to parametrize the intensity function as a weighted combination of random kernel functions, learn the parameters by maximum likelihood estimation, and eventually derive a sample complexity.

- (A1) $a(\mathcal{S}, t)$ is smooth and bounded on $[0, T]$: $0 < a_{\min} \leqslant a \leqslant a_{\max} < \infty$, and $\ddot{a} := d^2 a/dt^2$ is absolutely continuous with $\int \ddot{a}(t) dt < \infty$.
- (A2) There is a known distribution $\mathbb{Q}'(\boldsymbol{\tau})$ and a constant $C$ with $\mathbb{Q}'(\boldsymbol{\tau})/C \leqslant \mathbb{Q}(\boldsymbol{\tau}) \leqslant C\mathbb{Q}'(\boldsymbol{\tau})$.

**Kernel Smoothing**   To facilitate our finite dimensional parameterization, we first convolve the intensity function with $K(t) = k(t/\sigma)/\sigma$ where $\sigma$ is the bandwidth parameter and $k$ is a kernel function (such as the Gaussian RBF kernel $k(t) = e^{-t^2/2}/\sqrt{2\pi}$) with

$$0 \leqslant k(t) \leqslant \kappa_{\max}, \quad \int k(t)\, dt = 1, \quad \int t\, k(t)\, dt = 0, \quad \text{and} \quad \sigma_k^2 := \int t^2 k(t)\, dt < \infty. \quad (8)$$

The convolution results in a smoothed intensity $a^K(\mathcal{S}, t) = K(t) \star (df(\mathcal{S}, t)/dt) = d(K(t) \star \Lambda(\mathcal{S}, t))/dt$. By the property of convolution and exchanging derivative with integral, we have that

$$
\begin{aligned}
a^K(\mathcal{S}, t) &= d(Z \cdot \mathbb{E}_{\boldsymbol{\tau} \sim \mathbb{Q}(\boldsymbol{\tau})}[K(t) \star \phi(\boldsymbol{\chi}_{\mathcal{S}}^\top \boldsymbol{r}(t|\boldsymbol{\tau})]) / dt && \text{by definition of } f(\cdot) \\
&= Z \cdot \mathbb{E}_{\boldsymbol{\tau} \sim \mathbb{Q}(\boldsymbol{\tau})} \left[ d(K(t) \star \phi(\boldsymbol{\chi}_{\mathcal{S}}^\top \boldsymbol{r}(t|\boldsymbol{\tau})) / dt \right] && \text{exchange derivative and integral} \\
&= Z \cdot \mathbb{E}_{\boldsymbol{\tau} \sim \mathbb{Q}(\boldsymbol{\tau})} \left[ K(t) \star \delta(t - t(\mathcal{S}, \boldsymbol{r}) \right] && \text{by property of convolution and function } \phi(\cdot) \\
&= Z \cdot \mathbb{E}_{\boldsymbol{\tau} \sim \mathbb{Q}(\boldsymbol{\tau})} \left[ K(t - t(\mathcal{S}, \boldsymbol{\tau})) \right] && \text{by definition of } \delta(\cdot)
\end{aligned}
$$

where $t(\mathcal{S}, \boldsymbol{\tau})$ is the time when function $\phi(\boldsymbol{\chi}_{\mathcal{S}}^\top \boldsymbol{r}(t|\boldsymbol{\tau}))$ jumps from 0 to 1. If we choose small enough kernel bandwidth, $a^K$ only incurs a small bias from $a$. But the smoothed intensity still results in infinite number of parameters, due to the unknown distribution $\mathbb{Q}(\boldsymbol{\tau})$. To address this problem, we design the following random approximation with finite number of parameters.

**Random Function Approximation**   The key idea is to sample a collection of $W$ random change points $\boldsymbol{\tau}$ from a known distribution $\mathbb{Q}'(\boldsymbol{\tau})$ which can be different from $\mathbb{Q}(\boldsymbol{\tau})$. If $\mathbb{Q}'(\boldsymbol{\tau})$ is not very far way from $\mathbb{Q}(\boldsymbol{\tau})$, the random approximation will be close to $a^K$, and thus close to $a$. More specifically, we will denote the space of weighted combination of $W$ random kernel function by

$$\mathcal{A} = \left\{ a_{\boldsymbol{w}}^K(\mathcal{S}, t) = \sum_{i=1}^W w_i\, K(t - t(\mathcal{S}, \boldsymbol{\tau}_i)) \; : \; \boldsymbol{w} \geqslant 0, \frac{Z}{C} \leqslant \|\boldsymbol{w}\|_1 \leqslant ZC \right\}, \{\boldsymbol{\tau}_i\} \overset{i.i.d.}{\sim} \mathbb{Q}'(\boldsymbol{\tau}). \quad (9)$$

**Lemma 2.** *If $W = \tilde{O}(Z^2/(\epsilon\sigma)^2)$, then with probability $\geqslant 1 - \delta$, there exists an $\widetilde{a} \in \mathcal{A}$ such that* $\mathbb{E}_{\mathcal{S}} \mathbb{E}_t \left[ (a(\mathcal{S}, t) - \widetilde{a}(\mathcal{S}, t))^2 \right] := \mathbb{E}_{\mathcal{S} \sim \mathbb{P}(\mathcal{S})} \int_0^T \left[ (a(\mathcal{S}, t) - \widetilde{a}(\mathcal{S}, t))^2 \right] dt / T = O(\epsilon^2 + \sigma^4).$

The lemma then suggests to set the kernel bandwidth $\sigma = O(\sqrt{\epsilon})$ to get $O(\epsilon^2)$ approximation error.

## 5 Learning Algorithm

We develop a learning algorithm, referred to as TCOVERAGELEARNER, to estimate the parameters of $a_{\boldsymbol{w}}^K(\mathcal{S}, t)$ by maximizing the joint likelihood of all observed events based on convex optimization techniques as follows.

**Maximum Likelihood Estimation** Instead of directly estimating the time-varying coverage function, which is the cumulative intensity function of the counting process, we turn to estimate the intensity function $a(\mathcal{S}, t) = \partial\Lambda(\mathcal{S}, t)/\partial t$. Given $m$ *i.i.d.* counting processes, $\mathcal{D}^m := \{(\mathcal{S}_1, N_1(t)), \ldots, (\mathcal{S}_m, N_m(t))\}$ up to observation time $T$, the log-likelihood of the dataset is [11]

$$\ell(\mathcal{D}^m | a) = \sum_{i=1}^m \left\{ \int_0^T \{\log a(\mathcal{S}_i, t)\}\, dN_i(t) - \int_0^T a(\mathcal{S}_i, t)\, dt \right\}. \quad (10)$$

Maximizing the log-likelihood with respect to the intensity function $a(\mathcal{S}, t)$ then gives us the estimation $\widehat{a}(\mathcal{S}, t)$. The $W$-term random kernel function approximation reduces a function optimization problem to a finite dimensional optimization problem, while incurring only small bias in the estimated function.

---

**Algorithm 1** TCoverageLearner

---

INPUT : $\{(\mathcal{S}_i, N_i(t))\}, i = 1, \ldots, m;$
Sample $W$ random features $\tau_1, \ldots, \tau_W$ from $\mathbb{Q}'(\tau)$;
Compute $\{t(\mathcal{S}_i, \tau_w)\}, \{\boldsymbol{g}_i\}, \{\boldsymbol{k}(t_{ij})\}, i \in \{1, \ldots, m\}, w = 1, \ldots, W, t_{ij} < T;$
Initialize $\boldsymbol{w}^0 \in \Omega = \{\boldsymbol{w} \geqslant 0, \|\boldsymbol{w}\|_1 \leqslant 1\};$
Apply projected quasi-newton algorithm [12] to solve 11;
OUTPUT : $a_{\boldsymbol{w}}^K(\mathcal{S}, t) = \sum_{i=1}^{W} w_i K(t - t(\mathcal{S}, \boldsymbol{\tau}_i))$

---

**Convex Optimization**. By plugging the parametrization $a_{\boldsymbol{w}}^K(\mathcal{S}, t)$ (9) into the log-likelihood (10), we formulate the optimization problem as :

$$\min_{\boldsymbol{w}} \sum_{i=1}^{m} \left\{ \boldsymbol{w}^\top \boldsymbol{g}_i - \sum_{t_{ij} < T} \log \left( \boldsymbol{w}^\top \boldsymbol{k}(t_{ij}) \right) \right\} \quad \text{subject to} \quad \boldsymbol{w} \geqslant 0, \|\boldsymbol{w}\|_1 \leqslant 1, \quad (11)$$

where we define

$$\boldsymbol{g}_{ik} = \int_0^T K\left(t - t(\mathcal{S}_i, \boldsymbol{\tau}_k)\right) dt \quad \text{and} \quad \boldsymbol{k}_l(t_{ij}) = K(t_{ij} - t(\mathcal{S}_i, \boldsymbol{\tau}_l)), \quad (12)$$

$t_{ij}$ when the $j$-th event occurs in the $i$-th counting process. By treating the normalization constant $Z$ as a free variable which will be tuned by cross validation later, we simply require that $\|\boldsymbol{w}\|_1 \leqslant 1$. By applying the Gaussian RBF kernel, we can derive a closed form of $\boldsymbol{g}_{ik}$ and the gradient $\triangledown \ell$ as

$$\boldsymbol{g}_{ik} = \frac{1}{2} \left\{ \mathrm{erfc}\left( -\frac{t(\mathcal{S}_i, \boldsymbol{\tau}_k)}{\sqrt{2}h} \right) - \mathrm{erfc}\left( \frac{T - t(\mathcal{S}_i, \boldsymbol{\tau}_k)}{\sqrt{2}h} \right) \right\}, \triangledown \ell \quad = \sum_{i=1}^{m} \left\{ \boldsymbol{g}_i - \sum_{t_{ij} < T} \frac{\boldsymbol{k}(t_{ij})}{\boldsymbol{w}^\top \boldsymbol{k}(t_{ij})} \right\}.$$
$$(13)$$

A pleasing feature of this formulation is that it is convex in the argument $\boldsymbol{w}$, allowing us to apply various convex optimization techniques to solve the problem efficiently. Specifically, we first draw $W$ random features $\boldsymbol{\tau}_1, \ldots, \boldsymbol{\tau}_W$ from $\mathbb{Q}'(\boldsymbol{\tau})$. Then, we precompute the jumping time $t(\mathcal{S}_i, \boldsymbol{\tau}_w)$ for every source set $\{\mathcal{S}_i\}_{i=1}^{m}$ on each random feature $\{\boldsymbol{\tau}_w\}_{w=1}^{W}$. Because in general $|S_i| << n$, this computation costs $O(mW)$. Based on the achieved $m$-by-$W$ jumping-time matrix, we preprocess the feature vectors $\{\boldsymbol{g}_i\}_{i=1}^{m}$ and $\boldsymbol{k}(t_{ij}), i \in \{1, \ldots, m\}, t_{ij} < T$, which costs $O(mW)$ and $O(mLW)$ where $L$ is the maximum number of events caused by a particular source set before time $T$. Finally, we apply the projected quasi-newton algorithm [12] to find the weight $\boldsymbol{w}$ that minimizes the negative log-likelihood of observing the given event data. Because the evaluation of the objective function and the gradient, which costs $O(mLW)$, is much more expensive than the projection onto the convex constraint set, and $L << n$, the worst case computation complexity is thus $O(mnW)$. Algorithm 1 summarizes the above steps in the end.

**Sample Strategy**. One important constitution of our parametrization is to sample $W$ random change points $\boldsymbol{\tau}$ from a known distribution $\mathbb{Q}'(\boldsymbol{\tau})$. Because given a set $\mathcal{S}_i$, we can only observe the jumping time of the events in each counting process without knowing the identity of the covered items (which is a key difference from [8]), the best thing we can do is to sample from these historical data. Specifically, let the number of counting processes that a single item $s \in \mathcal{V}$ is involved to induce be $N_s$, and the collection of all the jumping timestamps before time $T$ be $\mathcal{J}_s$. Then, for the $s$-th entry of $\boldsymbol{\tau}$, with probability $|\mathcal{J}_s|/nN_s$, we uniformly draw a sample from $\mathcal{J}_s$; and with probability $1 - |\mathcal{J}_s|/nN_s$, we assign a time much greater than $T$ to indicate that the item will never be covered until infinity. Given the very limited information, although this $\mathbb{Q}'(\boldsymbol{\tau})$ might be quite different from $\mathbb{Q}(\boldsymbol{\tau})$, by drawing sufficiently large number of samples and adjusting the weights, we expect it still can lead to good results, as illustrated in our experiments later.

## 6   Sample Complexity

Suppose we use $W$ random features and $m$ training examples to compute an $\epsilon_\ell$-MLE solution $\widehat{a}$, *i.e.*,

$$\ell(\mathcal{D}^m | \widehat{a}) \geqslant \max_{a' \in \mathcal{A}} \ell(\mathcal{D}^m | a') - \epsilon_\ell.$$

The goal is to analyze how well the function $\widehat{f}$ induced by $\widehat{a}$ approximates the true function $f$. This sections describes the intuition and the complete proof is provided in the appendix.

A natural choice for connecting the error between $f$ and $\widehat{f}$ with the log-likelihood cost used in MLE is the Hellinger distance [22]. So it suffices to prove an upper bound on the Hellinger distance $h(a, \widehat{a})$ between $\widehat{a}$ and the true intensity $a$, for which we need to show a high probability bound on the (total) empirical Hellinger distance $\widehat{H}^2(a, a')$ between the two. Here, $h$ and $\widehat{H}$ are defined as

$$h^2(a, a') := \frac{1}{2}\mathbb{E}_{\mathcal{S}}\mathbb{E}_t \left[ \sqrt{a(\mathcal{S}, t)} - \sqrt{a'(\mathcal{S}, t)} \right]^2,$$

$$\widehat{H}^2(a, a') := \frac{1}{2}\sum_{i=1}^{m} \int_0^T \left[ \sqrt{a(\mathcal{S}_i, t)} - \sqrt{a'(\mathcal{S}_i, t)} \right]^2 dt.$$

The key for the analysis is to show that the empirical Hellinger distance can be bounded by a martingale plus some other additive error terms, which we then bound respectively. This martingale is defined based on our hypotheses and the martingales $M_i$ associated with the counting process $N_i$:

$$M(t|g) := \int_0^t g(t) d\left(\sum_i M_i(t)\right) = \sum_{i=1}^{m} \int_0^t g(t) dM_i(t)$$

where $g \in \mathcal{G} = \left\{ g_{a'} = \frac{1}{2}\log\frac{a+a'}{2a} : a' \in \mathcal{A} \right\}$. More precisely, we have the following lemma.

**Lemma 3.** *Suppose $\widehat{a}$ is an $\epsilon_\ell$-MLE. Then*

$$\widehat{H}^2\left(\widehat{a}, a\right) \leqslant 16M(T; g_{\widehat{a}}) + 4\left[ \ell(\mathcal{D}^m|a) - \max_{a' \in \mathcal{A}} \ell(\mathcal{D}^m|a') \right] + 4\epsilon_\ell.$$

The right hand side has three terms: the martingale (estimation error), the likelihood gap between the truth and the best one in our hypothesis class (approximation error), and the optimization error. We then focus on bounding the martingale and the likelihood gap.

To bound the martingale, we first introduce a notion called $(d, d')$-covering dimension measuring the complexity of the hypothesis class, generalizing that in [25]. Based on this notion, we prove a uniform convergence inequality, combining the ideas in classic works on MLE [25] and counting process [13]. Compared to the classic uniform inequality, our result is more general, and the complexity notion has more clear geometric interpretation and are thus easier to verify. For the likelihood gap, recall that by Lemma 2, there exists an good approximation $\tilde{a} \in \mathcal{A}$. The likelihood gap is then bounded by that between $a$ and $\tilde{a}$, which is small since $a$ and $\tilde{a}$ are close.

Combining the two leads to a bound on the Hellinger distance based on bounded dimension of the hypothesis class. We then show that the dimension of our specific hypothesis class is at most the number of random features $W$, and convert $\widehat{H}^2(\widehat{a}, a)$ to the desired $\ell_2$ error bound on $f$ and $\widehat{f}$.

**Theorem 4.** *Suppose $W = \tilde{O}\left( Z^2 \left[ \left(\frac{ZT}{\epsilon}\right)^{5/2} + \left(\frac{ZT}{\epsilon a_{\min}}\right)^{5/4} \right] \right)$ and $m = \tilde{O}\left( \frac{ZT}{\epsilon}[W + \epsilon_\ell] \right)$. Then with probability $\geqslant 1 - \delta$ over the random sample of $\{\tau_i\}_{i=1}^{W}$, we have that for any $0 \leqslant t \leqslant T$,*

$$\mathbb{E}_{\mathcal{S}} \left[ \widehat{f}(\mathcal{S}, t) - f(\mathcal{S}, t) \right]^2 \leqslant \epsilon.$$

The theorem shows that the number of random functions needed to achieve $\epsilon$ error is roughly $O(\epsilon^{-5/2})$, and the sample size is $O(\epsilon^{-7/2})$. They also depend on $a_{\min}$, which means with more random functions and data, we can deal with intensities with more extreme values. Finally, they increase with the time $T$, *i.e.*, it is more difficult to learn the function values at later time points.

## 7 Experiments

We evaluate TCOVERAGELEARNER on both synthetic and real world information diffusion data. We show that our method can be more robust to model misspecification than other state-of-the-art alternatives by learning a temporal coverage function all at once.

### 7.1 Competitors

Because our input data only include pairs of a source set and the temporal information of its triggered events $\{(\mathcal{S}_i, N_i(t))\}_{i=1}^{m}$ with unknown identity, we first choose the general kernel ridge regression model as the major baseline, which directly estimates the influence value of a source set

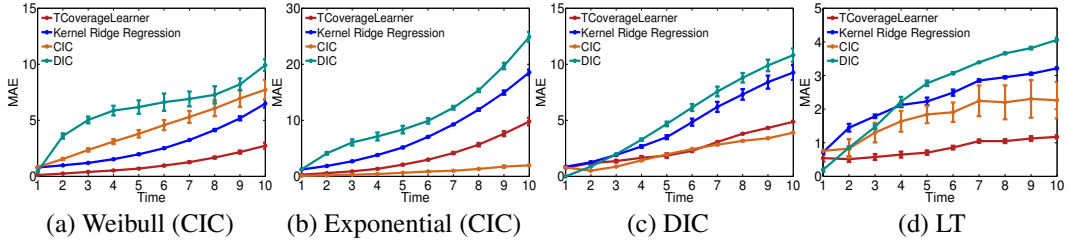

| (a) Weibull (CIC) | (b) Exponential (CIC) | (c) DIC | (d) LT |

Figure 1: MAE of the estimated influence on test data along time with the true diffusion model being continuous-time independent cascade with pairwise Weibull (a) and Exponential (b) transmission functions, (c) discrete-time independent cascade model and (d) linear-threshold cascade model.

$\chi_{\mathcal{S}}$ by $f(\chi_{\mathcal{S}}) = \boldsymbol{k}(\chi_{\mathcal{S}})(\mathbf{K} + \lambda\mathbf{I})^{-1}\mathbf{y}$ where $\boldsymbol{k}(\chi_{\mathcal{S}}) = K(\chi_{\mathcal{S}_i}, \chi_{\mathcal{S}})$, and $\mathbf{K}$ is the kernel matrix. We discretize the time into several steps and fit a separate model to each of them. Between two consecutive time steps, the predictions are simply interpolated. In addition, to further demonstrate the robustness of TCOVERAGELEARNER, we compare it to the two-stage methods which must know the *identity* of the nodes involved in an information diffusion process to first learn a specific diffusion model based on which they can then estimate the influence. We give them such an advantage and study three well-known diffusion models : (I) **C**ontinuous-time **I**ndependent **C**ascade model(CIC)[14, 15]; (II) **D**iscrete-time **I**ndependent **C**ascade model(DIC)[1]; and (III) **L**inear-**T**hreshold cascade model(LT)[1].

## 7.2  Influence Estimation on Synthetic Data

We generate Kronecker synthetic networks ([0.9 0.5;0.5 0.3]) which mimic real world information diffusion patterns [16]. For CIC, we use both Weibull distribution (Wbl) and Exponential distribution (Exp) for the pairwise transmission function associated with each edge, and randomly set their parameters to capture the heterogeneous temporal dynamics. Then, we use NETRATE [14] to learn the model by assuming an exponential pairwise transmission function. For DIC, we choose the pairwise infection probability uniformly from 0 to 1 and fit the model by [17]. For LT, we assign the edge weight $w_{uv}$ between $u$ and $v$ as $1/d_v$, where $d_v$ is the degree of node $v$ following [1]. Finally, 1,024 source sets are sampled with power-law distributed cardinality (with exponent 2.5), each of which induces eight independent cascades(or counting processes), and the test data contains another 128 independently sampled source sets with the ground truth influence estimated from 10,000 simulated cascades up to time $T = 10$. Figure 1 shows the MAE(Mean Absolute Error) between the estimated influence value and the true value up to the observation window $T = 10$. The average influence is 16.02, 36.93, 9.7 and 8.3. We use 8,192 random features and two-fold cross validation on the train data to tune the normalization $Z$, which has the best value 1130, 1160, 1020, and 1090, respectively. We choose the RBF kernel bandwidth $h = 1/\sqrt{2\pi}$ so that the magnitude of the smoothed approximate function still equals to 1 (or it can be tuned by cross-validation as well), which matches the original indicator function. For the kernel ridge regression, the RBF kernel bandwidth and the regularization $\lambda$ are all chosen by the same two-fold cross validation. For CIC and DIC, we learn the respective model up to time $T$ for once.

Figure 1 verifies that even though the underlying diffusion models can be dramatically different, the prediction performance of TCOVERAGELEARNER is robust to the model changes and consistently outperforms the nontrivial baseline significantly. In addition, even if CIC and DIC are provided with extra information, in Figure 1(a), because the ground-truth is continuous-time diffusion model with Weibull functions, they do not have good performance. CIC assumes the right model but the wrong family of transmission functions. In Figure 1(b), we expect CIC should have the best performance for that it assumes the correct diffusion model and transmission functions. Yet, TCOVERAGELEARNER still has comparable performance with even less information. In Figure 1(c), although DIC has assumed the correct model, it is hard to determine the correct step size to discretize the time line, and since we only learn the model once up to time $T$ (instead of at each time point), it is harder to fit the whole process. In Figure1(d), both CIC and DIC have the wrong model, so we have similar trend as Figure synthetic(a). Moreover, for kernel ridge regression, we have to first partition the timeline with arbitrary step size, fit the model to each of time, and interpolate the value between neighboring time legs. Not only will the errors from each stage be accumulated to the error of the final prediction, but also we cannot rely on this method to predict the influence of a source set beyond the observation window $T$.

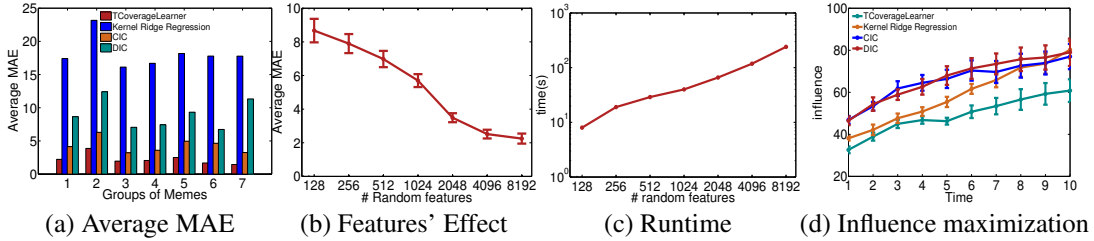

|   |   |   |   |
|---|---|---|---|
| (a) Average MAE | (b) Features' Effect | (c) Runtime | (d) Influence maximization |

Figure 2: (a) Average MAE from time 1 to 10 on seven groups of real cascade data; (b) Improved estimation with increasing number of random features; (c) Runtime in log-log scale; (d) Maximized influence of selected sources on the held-out testing data along time.

Overall, compared to the kernel ridge regression, TCOVERAGELEARNER only needs to be trained once given all the event data up to time $T$ in a compact and principle way, and then can be used to infer the influence of any given source set at any particular time much more efficiently and accurately. In contrast to the two-stage methods, TCOVERAGELEARNER is able to address the more general setting with much less assumption and information but still can produce consistently competitive performance.

## 7.3 Influence Estimation on Real Data

MemeTracker is a real-world dataset [18] to study information diffusion. The temporal flow of information was traced using quotes which are short textual phrases spreading through the websites. We have selected seven typical groups of cascades with the representative keywords like 'apple and jobs', 'tsunami earthquake', etc., among the top active 1,000 sites. Each set of cascades is split into 60%-train and 40%-test. Because we often can observe cascades only from single seed node, we rarely have cascades produced from multiple sources simultaneously. However, because our model can capture the correlation among multiple sources, we challenge TCOVERAGELEARNER with sets of randomly chosen multiple source nodes on the independent hold-out data. Although the generation of sets of multiple source nodes is simulated, the respective influence is calculated from the real test data as follows : Given a source set $\mathcal{S}$, for each node $u \in \mathcal{S}$, let $\mathcal{C}(u)$ denote the set of cascades generated from $u$ on the testing data. We uniformly sample cascades from $\mathcal{C}(u)$. The average length of all sampled cascades is treated as the true influence of $\mathcal{S}$. We draw 128 source sets and report the average MAE along time in Figure 2(a). Again, we can observe that TCOVERAGELEARNER has consistent and robust estimation performance across all testing groups. Figure 2(b) verifies that the prediction can be improved as more random features are exploited, because the representational power of TCOVERAGELEARNER increases to better approximate the unknown true coverage function. Figure 2(c) indicates that the runtime of TCOVERAGELEARNER is able to scale linearly with large number of random features. Finally, Figure 2(d) shows the application of the learned coverage function to the influence maximization problem along time, which seeks to find a set of source nodes that maximize the expected number of infected nodes by time $T$. The classic greedy algorithm[19] is applied to solve the problem, and the influence is calculated and averaged over the seven held-out test data. It shows that TCOVERAGELEARNER is very competitive to the two-stage methods with much less assumption. Because the greedy algorithm mainly depends on the relative rank of the selected sources, although the estimated influence value can be different, the selected set of sources could be similar, so the performance gap is not large.

## 8 Conclusions

We propose a new problem of learning temporal coverage functions with a novel parametrization connected with counting processes and develop an efficient algorithm which is guaranteed to learn such a combinatorial function from only polynomial number of training samples. Empirical study also verifies our method outperforms existing methods consistently and significantly.

**Acknowledgments**   This work was supported in part by NSF grants CCF-0953192, CCF-1451177, CCF-1101283, and CCF-1422910, ONR grant N00014-09-1-0751, AFOSR grant FA9550-09-1-0538, Raytheon Faculty Fellowship, NSF IIS1116886, NSF/NIH BIGDATA 1R01GM108341, NSF CAREER IIS1350983 and Facebook Graduate Fellowship 2014-2015.

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
