[Supplementary Material]

# A  Approximation Error

Recall that we view our data as a marked counting process

$$N_i(t) = f(\mathcal{S}_i, t) + M_i(t).$$

where $t \in [0, T]$ and $T$ is the time window, $\mathcal{S}_i \subseteq \mathcal{V}$ is the marker, and $M_i(t)$ is a zero mean local martingale.

We make the following assumptions for our analysis of the parametrization and estimation.

(A1)  $f(\mathcal{S}, t)$ has derivative $a(\mathcal{S}, t)$ with respect to $t$. For any $\mathcal{S}$, $a(\mathcal{S}, t) = df(\mathcal{S}, t)/dt$ is smooth and bounded on $[0, T]$: $a(\mathcal{S}, t)$ is smooth and bounded on $[0, T]$: $0 < a_{\min} \leqslant a \leqslant a_{\max} < \infty$, and $\ddot{a} := d^2 a/dt^2$ is absolutely continuous with $\int \ddot{a}(t) dt < \infty$.

(A2)  There is a known distribution $\mathbb{Q}'(\boldsymbol{\tau})$ and a constant $C$ with $\mathbb{Q}'(\boldsymbol{\tau})/C \leqslant \mathbb{Q}(\boldsymbol{\tau}) \leqslant C\mathbb{Q}'(\boldsymbol{\tau})$.

Let $a^K$ denote the convolution of $a$ with a kernel smoothing function $K$ with bandwidth $\sigma$. More precisely, $K(t) = \frac{1}{\sigma} k(\frac{t}{\sigma})$ and $k$ is a kernel with

$$0 \leqslant k(t) \leqslant \kappa_{\max}, \quad \int k(t)\, dt = 1, \quad \int t\, k(t)\, dt = 0, \quad \text{and} \quad \sigma_k^2 := \int t^2 k(t)\, dt < \infty.$$

Let

$$\mathcal{A} = \left\{ a_{\boldsymbol{w}}^K = \sum_{i=1}^{W} w_i K(t - t(\mathcal{S}_i, \boldsymbol{\tau}_i)) : \boldsymbol{w} \geqslant 0, \frac{Z}{C} \leqslant \|\boldsymbol{w}\|_1 \leqslant ZC \right\}$$

denote our hypothesis class. In the following, we show that there exists $\widetilde{a} \in \mathcal{A}$ that is close to $a$ when the number of features $W$ is sufficiently large. We first show that $a$ is close to $a^K$ and then show that there exists $\widetilde{a} \in \mathcal{A}$ close to $a^K$. The first step follows directly from a classic result in kernel density estimation.

**Lemma 5** (*e.g.*, Theorem 6.28 in [20]). *For any $\mathcal{S}$ and $t$, $a^K(\mathcal{S}, t) - a(\mathcal{S}, t) = O(\sigma^4)$.*

For the second step, we have the following lemma based on the quantitive $C$ measuring the difference between the true distribution $\mathbb{Q}$ of the features and the sample distribution $\mathbb{Q}'$.

**Lemma 6.** *Let $\mathbb{P}(\mathcal{S})$ be any distribution of $\mathcal{S}$. Suppose $\boldsymbol{\tau}_1, \ldots, \boldsymbol{\tau}_W$ are drawn i.i.d. from $\mathbb{Q}'(\boldsymbol{\tau})$, and $W = O\left( \left(\frac{CZ\kappa_{\max}}{\epsilon\sigma}\right)^2 \log \frac{1}{\delta\delta_1} \right)$. Then with probability at least $1 - \delta$ over $\boldsymbol{\tau}_1, \ldots, \boldsymbol{\tau}_W$, there exists $\widetilde{a} \in \mathcal{A}$ such that,*

$$\Pr_{\mathcal{S} \sim \mathbb{P}(\mathcal{S})} \left\{ \mathbb{E}_t \left[ \widetilde{a}(\mathcal{S}, t) - a^K(\mathcal{S}, t) \right]^2 \geqslant \epsilon^2 \right\} \leqslant \delta_1.$$

*Proof.* Let $a_i(\mathcal{S}, t) = Z \frac{\mathbb{Q}(\boldsymbol{\tau}_i)}{\mathbb{Q}'(\boldsymbol{\tau}_i)} K(t - t(\mathcal{S}, \boldsymbol{\tau}_i))$ for $i = 1, \ldots, W$. Then $\mathbb{E}_{\boldsymbol{\tau}_i \sim \mathbb{Q}'(\boldsymbol{\tau}_i)}[a_i] = a^K$. Let $\widetilde{a}(\mathcal{S}, t) = \frac{Z}{W} \sum_{i=1}^{W} \frac{\mathbb{Q}(\boldsymbol{\tau}_i)}{\mathbb{Q}'(\boldsymbol{\tau}_i)} K(t - t(\mathcal{S}, \boldsymbol{\tau}_i))$ be the sample average of these functions. Then $\widetilde{a} \in \mathcal{A}$ since $\frac{Z}{CW} \leqslant \frac{Z}{W} \frac{\mathbb{Q}(\boldsymbol{\tau}_i)}{\mathbb{Q}'(\boldsymbol{\tau}_i)} \leqslant \frac{ZC}{W}$.

Fix $\mathcal{S}$, and consider the Hilbert space with the inner product

$$\langle f, g \rangle = \mathbb{E}_t \left[ f(\mathcal{S}, t) g(\mathcal{S}, t) \right] = \frac{1}{T} \int_0^T f(\mathcal{S}, t) g(\mathcal{S}, t) dt.$$

We now apply the following lemma, which states that the average of bounded vectors in a Hilbert space concentrates towards its expectation in the Hilbert norm exponentially fast.

**Claim 1** (Lemma 4 in [21]). *Let $\boldsymbol{X} = \{x_1, \cdots, x_W\}$ be iid random variables in a ball $\mathcal{A}$ of radius $M$ centered around the origin in a Hilbert space. Denote their average by $\overline{\boldsymbol{X}} = \frac{1}{W} \sum_{i=1}^{W} x_i$. Then for any $\delta > 0$, with probability $1 - \delta$,*

$$\|\overline{\boldsymbol{X}} - \mathbb{E}\overline{\boldsymbol{X}}\| \leqslant \frac{M}{\sqrt{W}} \left( 1 + \sqrt{2 \log \frac{1}{\delta}} \right).$$

Since $\|\boldsymbol{w}\|_1 \leqslant CZ$ and $|K| \leqslant \frac{\kappa_{\max}}{\sigma}$, the norm $\|a_i\| \leqslant \frac{CZ\kappa_{\max}}{\sigma}$. Then when $W = O\left(\left(\frac{CZ\kappa_{\max}}{\epsilon\sigma}\right)^2 \log \frac{1}{\delta\delta_1}\right)$, for any fixed $\mathcal{S}$ we have

$$\Pr_{\boldsymbol{\tau}}\left[\mathbb{E}_t\left[\widetilde{a}(\mathcal{S},t) - a^K(\mathcal{S},t)\right]^2 \geqslant \epsilon^2\right] \leqslant \delta\delta_1$$

where $\Pr_{\boldsymbol{\tau}}$ is over the random sample of $\boldsymbol{\tau}_1, \ldots, \boldsymbol{\tau}_W$. This leads to

$$\Pr_{\mathcal{S}\sim\mathbb{P}(\mathcal{S})}\Pr_{\boldsymbol{\tau}}\left[\mathbb{E}_t\left[\widetilde{a}(\mathcal{S},t) - a^K(\mathcal{S},t)\right]^2 \geqslant \epsilon^2\right] \leqslant \delta\delta_1$$

Exchanging $\Pr_{\mathcal{S}\sim\mathbb{P}(\mathcal{S})}$ and $\Pr_{\boldsymbol{\tau}}$ by Fubini's theorem, and then by Markov's inequality, we have

$$\Pr_{\boldsymbol{\tau}}\left\{\Pr_{\mathcal{S}\sim\mathbb{P}(\mathcal{S})}\left[\mathbb{E}_t\left[\widetilde{a}(\mathcal{S},t) - a^K(\mathcal{S},t)\right]^2 \geqslant \epsilon^2\right] \geqslant \delta_1\right\} \leqslant \delta$$

This means with probability at least $1 - \delta$ over the random features, on at least $1 - \delta_1$ probability mass of the distribution of $\mathcal{S}$, $\mathbb{E}_t\left[\widetilde{a}(\mathcal{S},t) - a^K(\mathcal{S},t)\right]^2 \leqslant \epsilon^2$. □

Combining the two, we have the following approximation error bound.

**Lemma 2** *Let $\mathbb{P}(\mathcal{S})$ be any distribution of $\mathcal{S}$. Suppose $\boldsymbol{\tau}_1, \ldots, \boldsymbol{\tau}_W$ are drawn i.i.d. from $\mathbb{Q}'(\boldsymbol{\tau})$, and $W = O\left(\left(\frac{CZ\kappa_{\max}}{\epsilon\sigma}\right)^2 \log \frac{1}{\delta\delta_1}\right)$. Then with probability at least $1 - \delta$ over $\boldsymbol{\tau}_1, \ldots, \boldsymbol{\tau}_W$, there exists $\widetilde{a} \in \mathcal{A}$ such that with probability at least $1 - \delta_1$ over $\mathcal{S}$,*

$$\mathbb{E}_t\left[\widetilde{a}(\mathcal{S},t) - a(\mathcal{S},t)\right]^2 \leqslant \epsilon^2 + O(\sigma^4).$$

*Consequently, if $W = O\left(\left(\frac{CZ\kappa_{\max}}{\epsilon\sigma}\right)^2 \log \frac{a_{\max}+CZ\kappa_{\max}}{\delta\epsilon}\right)$, with probability at least $1 - \delta$ over $\boldsymbol{\tau}_1, \ldots, \boldsymbol{\tau}_W$, there exists $\widetilde{a} \in \mathcal{A}$ such that*

$$\mathbb{E}_{\mathcal{S}}\mathbb{E}_t\left[\widetilde{a}(\mathcal{S},t) - a(\mathcal{S},t)\right]^2 = O(\epsilon^2 + \sigma^4).$$

*Proof.* The first statement follows from Lemma 5 and 6. Since $\left[\widetilde{a}(\mathcal{S},t) - a(\mathcal{S},t)\right]^2 \leqslant C_1 := (a_{\max} + CZ\kappa_{\max})^2$, we can set $\delta_1 = \epsilon^2/C_1$. Then

$$\mathbb{E}_{\mathcal{S}}\mathbb{E}_t\left[\widetilde{a}(\mathcal{S},t) - a(\mathcal{S},t)\right]^2 \leqslant (1-\delta_1)(\epsilon^2 + O(\sigma^4)) + \delta_1 C_1 = O(\epsilon^2 + \sigma^4)$$

which completes the proof. □

For convenience, let $\epsilon_a^2 := O(\epsilon^2 + \sigma^4)$ denote the $\ell_2$ approximation error.

## B Sample Complexity

**Setup** Recall that the true intensity $a$ is bounded on $[0, T]$:

$$0 < a_{\min} \leqslant a \leqslant a_{\max} < \infty.$$

The kernel $K$ is also bounded on $[0, T]$:

$$0 < \kappa_{\min} \leqslant K(t) \leqslant \kappa_{\max}, \forall t \in [0, T]$$

where $\kappa_{\min} := \min_{t\in[0,T]} K(t) > 0$ is satisfied for typical kernels, *e.g.*, the Gaussian kernel. Our hypothesis class is

$$\mathcal{A} = \left\{a_{\boldsymbol{w}}^K = \sum_{i=1}^W w_i K(t - t(\mathcal{S}_i, \boldsymbol{\tau}_i)) : \boldsymbol{w} \geqslant 0, \frac{Z}{C} \leqslant \|\boldsymbol{w}\|_1 \leqslant ZC\right\}$$

and thus $a_{\boldsymbol{w}}^K$ is also bounded: $\forall a' \in \mathcal{A}$,

$$0 < a_{\min}^w := \frac{Z\kappa_{\min}}{C} \leqslant a'(\mathcal{S},t) \leqslant a_{\max}^w := CZ\kappa_{\max}, \forall \mathcal{S}, t \in [0, T].$$

With the exception of $\kappa_{\min}$ and $a_{\min}^w$ that depend on $\sigma$, all other parameters are treated as constants.

We observe $\mathcal{D}^m = \{(\mathcal{S}_i, N_i(t))\}_{i=1}^m$, and we want to fit $a(\mathcal{S}, t)$ by $a_{\boldsymbol{w}}^K(\mathcal{S}, t)$ by using maximum likelihood estimation (MLE). The log-likelihood is defined as

$$\ell(\mathcal{D}^m | a') := \sum_{i=1}^m \int_0^T \left[ \log a'(\mathcal{S}_i, t) \right] dN_i(t) - \sum_{i=1}^m \int_0^T a'(\mathcal{S}_i, t) dt$$

and we optimize the log-likelihood to find an approximate solution.

**Definition 7.** *We say that $\widehat{a} \in \mathcal{A}$ is an $\epsilon_\ell$-MLE if*

$$\ell(\mathcal{D}^m | \widehat{a}) \geqslant \max_{a' \in \mathcal{A}} \ell(\mathcal{D}^m | a') - \epsilon_\ell.$$

**Analysis Roadmap**  Our final goal is to bound the $\ell_2$ error between the truth $f(t)$ and the function $\widehat{f}(t) = \int_0^t \widehat{a}(s) ds$ induced by the MLE output $\widehat{a}$. A natural choice for connecting $\ell_2$ error with the log-likelihood cost used in MLE is the Hellinger distance. So it suffices to prove an upper bound on the hellinger distance between the MLE output $\widehat{a}$ and the truth $a$, for which we need to show a high probability bound on the empirical Hellinger distance between the two. The key for the analysis is to show that the empirical Hellinger distance can be bounded by a martingale plus some additive error terms. This martingale is defined based on the martingales $M_i$ associated with the counting process $N_i$. The additive error terms are the optimization error and the likelihood gap between the truth and the best one in our hypothesis class. Therefore, our analysis focuses on two parts: a high probability bound for the martingale, and a high probability bound on the likelihood gap.

To bound the martingale, we need to show a uniform convergence inequality. We first introduce a dimension notion measuring the complexity of the hypothesis class, and then prove the uniform convergence based on this notion. Compared to classic uniform inequality for (unmarked) counting process [13], our uniform inequality is for marked counting processes, and the complexity notion and the related conditions have more clear geometric interpretation and are thus easier to verify.

To bound the likelihood gap, we decompose it into three terms, related respectively to the martingale part of the counting processes, the compensate part of the counting processes, and the cumulative difference between the two intensities $a$ and $\widehat{a}$. The first term can be bounded by bounding its variance and applying a classic martingale inequality. The second term reduces to the KL-divergence, which can be bounded by the $\ell_2$ approximation error between the truth and the hypotheses. Similarly, the cumulative difference between the two intensities can be bounded by the $\ell_2$ approximation error.

We then combine the two to get a bound on the Hellinger distance between the MLE output and the truth based on the dimension of the hypothesis class. This bound is for general hypothesis class, so we bound the dimension of our specific hypothesis class. Finally, we convert the Hellinger distance between the MLE output and the truth to the desired $\ell_2$ error bound on $f$ and $\widehat{f}$.

The rest of the section is organized as follows. We first show the construction of the key martingale upper bound for the Hellinger distance in Section B.1, and then show how to bound the martingale and the likelihood gap in Section B.2 and Section B.3 respectively. In Section B.4 we provide the general bound for the Hellinger distance based on the dimension of the hypothesis class. Finally, in Section B.5 we bound the dimension of our hypothesis class and convert the Hellinger distance to $\ell_2$ error, achieving the final bound for learning time varying coverage functions.

## B.1  Upper Bound the Hellinger Distance

More precisely, the Hellinger distance is defined as

$$h^2(a, a') = \frac{1}{2} \mathbb{E}_{\mathcal{S}} \mathbb{E}_t \left[ \sqrt{a(\mathcal{S}, t)} - \sqrt{a'(\mathcal{S}, t)} \right]^2$$

where $\mathbb{E}_{\mathcal{S}}$ is with respect to the random drawing of $\mathcal{S}$, and $\mathbb{E}_t \left[ g(t) \right]$ denotes $\frac{1}{T} \int_0^T g(t) dt$. Define the (total) empirical Hellinger distance as

$$\widehat{H}^2(a, a') = \frac{1}{2} \sum_{i=1}^m \int_0^T \left[ \sqrt{a(\mathcal{S}_i, t)} - \sqrt{a'(\mathcal{S}_i, t)} \right]^2 dt$$

and note that $\mathbb{E}_{\mathcal{S}} \mathbb{E}_t \left[ \widehat{H}^2(a, a') \right] = mTh^2(a, a')$.

Define a martingale

$$M(t|g) := \int_0^t g(t)d\left(\sum_i M_i(t)\right) = \sum_{i=1}^m \int_0^t g(t)dM_i(t) \tag{14}$$

where $M_i(t)$ is the martingale in the marked counting process $(\mathcal{S}_i, N_i(t))$, and $g \in \mathcal{G}$ where $\mathcal{G}$ is a set of functions defined as

$$\mathcal{G} = \left\{ g_{a'} = \frac{1}{2}\log\frac{a+a'}{2a} : a' \in \mathcal{A} \right\}.$$

Let $V_n(t|g)$ denote the $n$-th order variation process corresponding to $M(t|g)$.

Define two distances on $\mathcal{G}$:

$$d_{2,t}^2(g,g') = \frac{1}{2}\sum_{i=1}^m \int_0^t \left[\exp(g) - \exp(g')\right]^2 d\Lambda_i(t)$$

where $\Lambda_i(t) = f(\mathcal{S}_i, t)$ is the compensator of $N_i(t)$ and

$$d_{\infty,t}(g,g') = \max_{\tau \in [0,t], \mathcal{S}} |\exp(g(\mathcal{S},\tau)) - \exp(g'(\mathcal{S},\tau))|.$$

Now we show that $\widehat{H}(\cdot,\cdot)$ can be bounded by a martingale plus some additive error terms.

**Lemma 3** *Suppose $\widehat{a}$ is an $\epsilon_\ell$-MLE. Then*

$$\widehat{H}^2\left(\frac{\widehat{a}+a}{2},a\right) \leqslant M(T|g_{\widehat{a}}) + \frac{1}{4}\left[\ell(\mathcal{D}^m|a) - \max_{a'\in\mathcal{A}}\ell(\mathcal{D}^m|a')\right] + \frac{1}{4}\epsilon_\ell,$$

$$\widehat{H}^2(\widehat{a},a) \leqslant 16M(T|g_{\widehat{a}}) + 4\left[\ell(\mathcal{D}^m|a) - \max_{a'\in\mathcal{A}}\ell(\mathcal{D}^m|a')\right] + 4\epsilon_\ell.$$

*Proof.* This is a generalization of Lemma 4.1 in [13] and the proof largely follows their arguments.

**Claim 2.** *For any $b \geqslant 0$,*

$$\frac{1}{2}[\ell(\mathcal{D}^m|b) - \ell(\mathcal{D}^m|a)] \leqslant M\left(T\left|\frac{1}{2}\log\frac{b}{a}\right.\right) - \widehat{H}^2(b,a).$$

*Proof.* Let $h_b := \frac{1}{2}\log\frac{b}{a}$.

$$\frac{1}{2}[\ell(\mathcal{D}^m|b) - \ell(\mathcal{D}^m|a)] = M(T|h_b) + \sum_{i=1}^m \int_0^T h_b d\Lambda_i(t) - \frac{1}{2}\sum_{i=1}^m \int_0^T (b-a)dt$$

and

$$
\begin{aligned}
\sum_{i=1}^m \int_0^T h_b d\Lambda_i(t) - \frac{1}{2}\sum_{i=1}^m \int_0^T (b-a)dt &= \sum_{i=1}^m \int_0^T \log\sqrt{\frac{b}{a}}\,d\Lambda_i(t) - \frac{1}{2}\sum_{i=1}^m \int_0^T (b-a)dt \\
&\leqslant \sum_{i=1}^m \int_0^T \left(\sqrt{\frac{b}{a}}-1\right)d\Lambda_i(t) - \frac{1}{2}\sum_{i=1}^m \int_0^T (b-a)dt \\
&= \sum_{i=1}^m \int_0^T \left(\sqrt{ba}-a\right)dt - \frac{1}{2}\sum_{i=1}^m \int_0^T (b-a)dt \\
&= -\widehat{H}^2(b,a).
\end{aligned}
$$

This completes the proof. $\square$

**Claim 3.** *For any $\widehat{a} \geqslant 0$,*

$$\ell\left(\mathcal{D}^m\left|\frac{\widehat{a}+a}{2}\right.\right) - \ell(\mathcal{D}^m|a) \geqslant \frac{1}{2}[\ell(\mathcal{D}^m|\widehat{a}) - \ell(\mathcal{D}^m|a)].$$

*Proof.* By the concavity of the log function,

$$
\begin{aligned}
\ell\left(\mathcal{D}^m\Big|\frac{\widehat{a}+a}{2}\right) - \ell(\mathcal{D}^m|a) &= \sum_{i=1}^m \int_0^T \log\left(\frac{\widehat{a}+a}{2a}\right) dN_i(t) - \sum_{i=1}^m \int_0^T \left(\frac{\widehat{a}+a}{2} - a\right) dt \\
&\geqslant \frac{1}{2}\sum_{i=1}^m \int_0^T \log\left(\frac{\widehat{a}}{a}\right) dN_i(t) - \frac{1}{2}\sum_{i=1}^m \int_0^T (\widehat{a}-a)dt \\
&= \frac{1}{2}[\ell(\mathcal{D}^m|\widehat{a}) - \ell(\mathcal{D}^m|a)]. \qquad (15)
\end{aligned}
$$

$\square$

We let $b = \frac{\widehat{a}+a}{2}$ in Claim 2 and combine with Claim 3, which then leads to

$$
\frac{1}{2}[\ell(\mathcal{D}^m|\widehat{a}) - \ell(\mathcal{D}^m|a)] \leqslant M\left(T\Big|\frac{1}{2}\log\frac{\widehat{a}+a}{2a}\right) - \widehat{H}^2(b,a).
$$

Note that $\frac{1}{2}\log\frac{\widehat{a}+a}{2a}$ is just $g_{\widehat{a}}$. This, together with the definition of $\epsilon_\ell$-MLE, completes the proof for the first statement.

For the second statement, we use the following claim.

**Claim 4** ([22]). $2\widehat{H}^2(\frac{a+b}{2},a) \leqslant \widehat{H}^2(b,a) \leqslant 16\widehat{H}^2(\frac{a+b}{2},a)$.

The second statement then follows from the first statement. $\qquad\square$

## B.2 Bounding the Martingale

We begin with some basics about martingales. Here, for a martingale $M(t)$, let $V_n(t)$ denote its $n$-th order variation process for $n \geqslant 2$, and let $V(t) := V_2(t)$. In particular,

$$
V(t) := \lim_{j\to\infty} \sum_{k=1}^n \operatorname{Var}(\Delta M_k \,|\, \mathcal{H}_{(k-1)t/j}) \qquad (16)
$$

where the time interval $[0,t]$ is partitioned into $j$ subintervals each of length $t/j$, and $\Delta M_k := M(kt/j) - M((k-1)t/j)$ is the increment of the martingale over the $k$th of these intervals. The higher order moments are defined similarly.

Informally, the increment $dV(t)$ of the predictable variation process can be written as $dV(t) = \operatorname{Var}(dM(t)\,|\,\mathcal{H}_{t-}) = \operatorname{Var}(dN(t)\,|\,\mathcal{H}_{t-})$, since $a(t)$ is predictable given $\mathcal{H}_{t-}$. Finally, $dN(t)$ may only take the value 0 or 1, and it follows that $dV(t) = a(t)dt(1 - a(t)dt) \approx a(t)dt = d\Lambda(t)$. This motivates the following claim, which will be useful in our later analysis.

**Claim 5** ( [11]). $V(t) = \int_0^t a(s)\,ds = \Lambda(t)$.

The following two classic martingale inequalities will also be useful.

**Lemma 8** ([23]). *Suppose that $|dM(t)| \leqslant C_M$ for all $t \geqslant 0$ and some $0 \leqslant C_M < \infty$, and let $V(t)$ denote its variation process. Then for each $x > 0$, $y > 0$,*

$$
\Pr\left[M(t) \geqslant x \text{ and } V(t) \leqslant y^2 \text{ for some } t\right] < \exp\left[-\frac{x^2}{2(xC_M + y^2)}\right].
$$

**Lemma 9** ([13]). *Suppose for all $t \geqslant 0$ and some constant $0 < C_M < \infty$,*

$$
V_n(t) \leqslant \frac{n!}{2}C_M^{n-2}R(t), \quad \forall n \geqslant 2,
$$

*where $R(t)$ is a predictable process. Then for each $x > 0$, $y > 0$,*

$$
\Pr\left[M(t) \geqslant x \text{ and } R(t) \leqslant y^2 \text{ for some } t\right] < \exp\left[-\frac{x^2}{2(xC_M + y^2)}\right].
$$

**Uniform Inequality for Marked Counting Processes** Now, we will prove a uniform inequality for the martingale $M(t|g)$ defined in (14), which is based on the marked counting process and the function $g \in \mathcal{G}$. Consider the following complexity notion for $\mathcal{G}$ based on a covering argument.

**Definition 10.** *Suppose $d$ and $d'$ are two families of metrics on $\mathcal{G}$ which are indexed by $t$, that is, for any $t \geqslant 0$, $d_t$ and $d'_t$ are two metrics on $\mathcal{G}$. The $(d, d')$-covering dimension of $\mathcal{G}$ is the minimum $D \geqslant 1$ such that there exist $c_1 \geqslant 1$ and $c_2 \geqslant 1$ satisfying the following. For each $\epsilon > 0$ and each ball $\mathcal{B} \subseteq \mathcal{G}$ with radius $R \geqslant \epsilon$, one can find $\mathcal{C} \subseteq \mathcal{G}$ with*

$$|\mathcal{C}| \leqslant (c_1 R / \epsilon)^D$$

*that is an $\epsilon$-covering of $\mathcal{B}$ for the $d_t$ metric and a $(c_2 \epsilon)$-covering for the $d'_t$ metric for each $t \geqslant 0$.*

Based on this notion we have the following uniform inequality.

**Theorem 11.** *Let $D$ be the $(d, d')$-covering dimension of $\mathcal{G}$. Suppose for any $g, g' \in \mathcal{G}$, any $n \geqslant 2$,*

$$V_n(t|g - g') \leqslant C_1 \frac{n!}{2} C_2^{n-2} d_t^2(g, g'),$$

*and*

$$V_n(t|g - g') \leqslant C_3 \frac{n!}{2} [C_4 d'_t(g, g')]^{n-2} d_t^2(g, g')$$

*where $C_1, C_2, C_3, C_4 > 0$ are some constants. Then there exists a constant $C_0 > 0$, such that for any $g^* \in \mathcal{G}$, any $y, z > 0$ and $x \geqslant C_0(y + 1)(z + D)$,*

$$\Pr\left[M(t|g - g^*) \geqslant x \text{ and } d_t(g^*, g) \leqslant y \text{ for some } t \text{ and some } g \in \mathcal{G}\right] \leqslant \exp\left[-z\right].$$

**Corollary 12.** *Let $D, V_n$ as specified in Theorem 11. Then there exists a constant $C_0 > 0$, such that for any $g^* \in \mathcal{G}$, any $y > 0$ and $0 < \delta < 1$, we have that with probability $\geqslant 1 - \delta$,*

$$M(t|g - g^*) \leqslant C_0(y + 1)\left(D + \log \frac{1}{\delta}\right)$$

*for all $g$ and $t$ satisfying $d_t(g^*, g) \leqslant y$.*

*Proof.* Let $M(\cdot)$ denote $M(t|\cdot)$ for short. For each $k = 0, 1, 2, \ldots$, for the ball $\mathcal{B}(g^*, y)$ and $\delta_k = O(2^{-k})y$, there exists a subset $\mathcal{C}_k$ of size $\exp\{O(kD)\}$ that is both a $\delta_k$-covering with respect to $d_t$ and a $(r\delta_k)$-covering with respect to $d'_t$ for some constant $r > 0$. Let $g_k$ denote the one in $\mathcal{C}_k$ closest to $g$. Since $g = g_0 + \sum_{k=0}^{\infty}(g_{k+1} - g_k)$, we have

$$\Pr\left[M(g - g^*) \geqslant x \text{ and } d_t(g^*, g) \leqslant y \text{ for some } t \text{ and some } g \in \mathcal{G}\right]$$

$$\leqslant \sum_{g_0 \in \mathcal{C}_0} \Pr\left[M(g_0 - g^*) > \eta \text{ and } d_t(g_0, g^*) \leqslant 2y \text{ for some } t\right]$$

$$+ \sum_{k=0}^{\infty} \sum_{g_k, g_{k+1}} \Pr\left[M(g_k - g_{k+1}) > \eta_k \text{ and } d_t(g_k, g_{k+1}) \leqslant 2\delta_k \text{ for some } t\right]$$

as long as $\eta + \sum_{k=0}^{\infty} \eta_k \leqslant x$.

We have by Lemma 9 that

$$\Pr\left[M(g_0 - g^*) > \eta \text{ and } d_t(g_0, g^*) \leqslant 2y \text{ for some } t\right] \leqslant \exp\left[-O\left(\frac{\eta^2}{\eta + y^2}\right)\right].$$

Also, for $g_k, g_{k+1}$ we have

$$V_n(t|g_k - g_{k+1}) \leqslant C_3 \frac{n!}{2} d_t^2(g_k, g_{k+1})[C_4 d'_t(g_k, g_{k+1})]^{n-2}$$

$$\leqslant C_3 \frac{n!}{2} d_t^2(g_k, g_{k+1})[C_4 d'_t(g_k, g) + C_4 d'_t(g, g_{k+1})]^{n-2}$$

$$\leqslant C_3 \frac{n!}{2} d_t^2(g_k, g_{k+1})[C_4 r\delta_k + C_4 r\delta_{k+1}]^{n-2}$$

$$\leqslant C_3 \frac{n!}{2} d_t^2(g_k, g_{k+1}) [2C_4 r\delta_k]^{n-2}.$$

Then by Lemma 9,

$$\Pr\left[M(g_k - g_{k+1}) > \eta_k \text{ and } d_t(g_k, g_{k+1}) \leqslant 2\delta_k \text{ for some } t\right] \leqslant \exp\left[-O\left(\frac{\eta_k^2}{\eta_k \delta_k + \delta_k^2}\right)\right].$$

Note that $\eta = O(y\sqrt{z} + cz)$ ensures $\frac{\eta^2}{c\eta+y^2} \geqslant z$. So we can choose $\eta = O(y\sqrt{z+D}+z+D)$ and $\eta_k = O(\delta_k(z+kD))$ so that the final statement holds.

We still need to verify $\eta + \sum_{k=0}^{\infty} \eta_k \leqslant x$. Since $\eta = O(y\sqrt{z+D}+z+D)$ and $\eta_k = O(\delta_k(z+kD)) = O(2^{-k}y(z+kD))$, it suffices to have $x = O((y+1)(z+D))$. $\qquad\square$

## B.3 Bounding the Likelihood Gap

**Lemma 13.** *Suppose there exists an $\widetilde{a} \in \mathcal{A}$ such that with probability at least $1 - \delta_1$ over $\mathcal{S}$, $\mathbb{E}_t\left[a'(\mathcal{S},t) - a(\mathcal{S},t)\right]^2 \leqslant \epsilon_a^2$. With probability $\geqslant 1 - m\delta_1$ over $\{\mathcal{S}_i\}_{i=1}^m$, we have that with probability $\geqslant 1 - \delta_2$ over $\{M_i\}_{i=1}^m$,*

$$\ell(\mathcal{D}^m|a) - \ell(\mathcal{D}^m|\widetilde{a}) \leqslant B(\delta_2) := O\left(\sqrt{c_\ell^2 Q \log \frac{1}{\delta_2}} + \log\frac{1}{\delta_2} + Q\log\frac{a_{\max}}{a_{\min}^w} + mT\epsilon_a\right)$$

*where*

$$Q = \frac{mT\epsilon_a^2}{a_{\min} + a_{\min}^w}, \quad \text{and} \quad c_\ell^2 = \frac{4\left(\sqrt{\frac{a_{\max}}{a_{\min}^w}} - 1 - \frac{1}{2}\log\left(\frac{a_{\min}^w}{a_{\max}}\right)\right)}{\left(\sqrt{\frac{a_{\min}^w}{a_{\max}}} - 1\right)^2}.$$

**Corollary 14.** *Under the condition of Lemma 13, $\ell(\mathcal{D}^m|a) - \max_{a' \in \mathcal{A}} \ell(\mathcal{D}^m|a') \leqslant B(\delta_2)$.*

*Proof.* With probability $\geqslant 1 - m\delta_1$, $\mathbb{E}_t\left[a(\mathcal{S}_i,t) - a(\mathcal{S}_i,t)\right]^2 \leqslant \epsilon_a^2$ for all $\mathcal{S}_i$. Assume this is true.

$$\ell(\mathcal{D}^m|a) - \ell(\mathcal{D}^m|\widetilde{a})$$

$$= \sum_{i=1}^m \left[\int_0^T (\log a - \log\widetilde{a})dN_i(t) - \int_0^T (a - \widetilde{a})dt\right]$$

$$= \sum_{i=1}^m \left[\underbrace{\int_0^T \log\left(\frac{a}{\widetilde{a}}\right)dM_i(t)}_{T_{i1}} + \underbrace{\int_0^T \log\left(\frac{a}{\widetilde{a}}\right)d\Lambda_i(t)}_{T_{i2}} - \underbrace{\int_0^T (a-\widetilde{a})dt}_{T_{i3}}\right]$$

where $\Lambda_i(t) := f(\mathcal{S}_i,t)$ is the compensator of $N_i(t)$. There are three terms under the sum, each of which is bounded in the following.

**Bounding $T_{i1}$** The first term $T_{i1}$ has zero expectation, and its variance is $\mathrm{Var}(T_{i1}) = \mathbb{E}_M\left[T_{i1}^2\right]$. Then

$$\mathbb{E}_M\left[T_{i1}^2\right] = \int_0^T \log^2\left(\frac{\widetilde{a}}{a}\right)dV_i(t) = \int_0^T \log^2\left(\frac{\widetilde{a}}{a}\right)d\Lambda_i(t) = 4\int_0^T \left[\frac{1}{2}\log\left(\frac{\widetilde{a}}{a}\right)\right]^2 d\Lambda_i(t).$$

We now apply the following claim:

**Claim 6** ([24])**.** *If $g \geqslant -L$ for some constant $L > 0$, then*

$$|g|^n \leqslant \frac{n!}{2}C_L^2\frac{1}{2}\left[\exp(g) - 1\right]^2, \text{ for any } n \geqslant 2,$$

*where $C_L^2 = \frac{4(e^L-1-L)}{(e^{-L}-1)^2}$.*

Since $\frac{1}{2}\log\left(\frac{\widetilde{a}}{a}\right) \geqslant \frac{1}{2}\log\left(\frac{a_{\min}^w}{a_{\max}}\right)$, by the above claim we have

$$\left[\frac{1}{2}\log\left(\frac{\widetilde{a}}{a}\right)\right]^2 \leqslant O(c_\ell^2)\left(\sqrt{\frac{\widetilde{a}}{a}} - 1\right)^2$$

and thus

$$
\begin{aligned}
\mathbb{E}_M\left[T_{i1}^2\right] &\leqslant O(c_\ell^2) \int_0^T a\left(\sqrt{\frac{\widetilde{a}}{a}} - 1\right)^2 dt = O(c_\ell^2) \int_0^T \left(\sqrt{\widetilde{a}} - \sqrt{a}\right)^2 dt \\
&\leqslant O(c_\ell^2 T)\mathbb{E}_t\left(\frac{\widetilde{a} - a}{\sqrt{\widetilde{a}} + \sqrt{a}}\right)^2 \\
&\leqslant B_1 := O\left(\frac{c_\ell^2 T \epsilon_a^2}{a_{\min} + a_{\min}^w}\right).
\end{aligned}
$$

We have that the variance of $\sum_i T_{i1}$ is bounded by $mB_1$ and that $|\sum_i dM(t|\mathcal{S}_i)| \leqslant 1$ almost surely. By martingale inequality in Lemma 8,

$$
\Pr_M\left[\sum_i T_{i1} \geqslant C_1\left(\sqrt{mB_1 \log\frac{1}{\delta_2}} + \log\frac{1}{\delta_2}\right)\right] \leqslant \frac{\delta_2}{2}
$$

for sufficiently large $C_1$.

**Bounding $T_{i2}$**   Since $T_{i2}$ is just the KL-divergence between $a(\mathcal{S}_i, \cdot)$ and $\widetilde{a}(\mathcal{S}_i, \cdot)$, we can apply the following claim.

**Claim 7** (Eqn (7.6) in Lemm 5 in [25]). *The KL-divergence between $g(\cdot)$ and $\widetilde{g}(\cdot)$ is at most $4 + 2\log\left[\max_t \left|\frac{g(t)}{\widetilde{g}(t)}\right|\right]$ times their Hellinger distance $\frac{1}{2}\int_0^T(\sqrt{g(t)} - \sqrt{\widetilde{g}(t)})dt$.*

By this claim,we have

$$
\begin{aligned}
\int_0^T \log\left(\frac{a}{\widetilde{a}}\right) d\Lambda_i(t) &\leqslant \left(4 + 2\log\left[\max_t\left|\sqrt{\frac{a(\mathcal{S}_i, t)}{\widetilde{a}(\mathcal{S}_i, t)}}\right|\right]\right)\int_0^T \left(\sqrt{a(\mathcal{S}_i, t)} - \sqrt{\widetilde{a}(\mathcal{S}_i, t)}\right)^2 dt \\
&\leqslant \left(4 + 2\log\frac{a_{\max}}{a_{\min}^w}\right)\int_0^T \left(\frac{a(\mathcal{S}_i, t) - \widetilde{a}(\mathcal{S}_i, t)}{\sqrt{a(\mathcal{S}_i, t)} + \sqrt{\widetilde{a}(\mathcal{S}_i, t)}}\right)^2 dt \\
&\leqslant B_2 := \left(4 + 2\log\frac{a_{\max}}{a_{\min}^w}\right)\frac{T\epsilon_a^2}{a_{\min} + a_{\min}^w}.
\end{aligned}
$$

**Bounding $T_{i3}$**   For $T_{i3}$, we have

$$
|T_{i3}| \leqslant \int_0^T |a(\mathcal{S}_i, t) - \widetilde{a}(\mathcal{S}_i, t)|dt \leqslant T\sqrt{\mathbb{E}_t|a(\mathcal{S}_i, t) - \widetilde{a}(\mathcal{S}_i, t)|^2} = T\epsilon_a =: B_3.
$$

Combining the bounds together, we have that $\ell(\mathcal{D}^m|a) - \max_{a' \in \mathcal{A}}\ell(\mathcal{D}^m|a')$ is bounded by $O\left(\sqrt{B_1 \log\frac{1}{\delta_2}} + \log\frac{1}{\delta_2} + m(B_2 + B_3)\right)$. □

### B.4   MLE for Marked Counting Processes

Here we apply Theorem 11 to bound the empirical Hellinger distance between an approximate MLE and the truth.

**Theorem 15.** *Let $D$ be the $(d_2, d_\infty)$-covering dimension of $\mathcal{G}$, and $\widehat{a}$ be an $\epsilon_\ell$-MLE. There exist constants $C_1, C_2 > 1$ such that for any $\{\mathcal{S}_i\}_{i=1}^m$, if $z \geqslant C_1[D + \Delta + \epsilon_\ell]$, then we have*

$$
\Pr_M\left[\widehat{H}^2(\widehat{a}, a) \geqslant z\right] \leqslant \exp\left[-z/C_2\right] + \Pr_M\left[\ell(\mathcal{D}^m|a) - \max_{a' \in \mathcal{A}}\ell(\mathcal{D}^m|a') > \Delta\right]
$$

*where $\Pr_M$ is with respect to the randomness in $\{M_i\}_{i=1}^m$.*

*Proof.* We first verify the conditions of Theorem 11 is satisfied and then apply it to prove the claim.

Since $g, g' \in \mathcal{G}$ are lower bounded by $\frac{1}{2}\log\frac{1}{2}$, Claim 6 leads to

$$|g - g'|^n \leqslant C_1' \frac{n!}{2}\frac{1}{2}\left[\exp(g) - \exp(g')\right]^2, \text{ for any } n \geqslant 2$$

for some constant $C_1' > 0$. Since $(\mathcal{S}_i, N_i(t))$ are independent, and the counting process $|dM_i(t)| \leqslant C_2' = 1$ for all $t$ and $\mathcal{S}$, then

$$
\begin{aligned}
V_n(t|g - g') &= \sum_i \int_0^t |g - g'|^n dV_{i,n} \\
&\leqslant (C_2')^{n-2} \sum_i \int_0^t |g - g'|^n dV_{i,2} = (C_2')^{n-2} \sum_i \int_0^t |g - g'|^n d\Lambda_i \\
&\leqslant C_1'(C_2')^{n-2}\frac{n!}{2} d_t^2(g, g')
\end{aligned}
$$

where $V_{i,n}$ are the $n$-th order variation processes for $M_i$, and $\Lambda_i$ is the compensator of $M_i$. This verifies the first condition. For the second condition, by Claim 6 we have

$$|g(\mathcal{S}, t) - g'(\mathcal{S}, t)|^2 \leqslant C_1'\frac{2!}{2}\frac{1}{2}\left[\exp(g(\mathcal{S}, t)) - \exp(g'(\mathcal{S}, t))\right]^2 \leqslant C_4^2 d_{\infty, t}^2(g, g')$$

where $(C_4')^2 = C_1'\frac{2!}{2}\frac{1}{2}$. Then

$$|g(\mathcal{S}, t) - g'(\mathcal{S}, t)|^{n-2} = (|g(\mathcal{S}, t) - g'(\mathcal{S}, t)|^2)^{(n-2)/2} \leqslant [C_4' d_{\infty, t}(g, g')]^{n-2}$$

and

$$
\begin{aligned}
V_n(t|g - g') &= \sum_i \int_0^t |g - g'|^n dV_{i,n} \leqslant (C_2')^{n-2} \sum_i \int_0^t |g - g'|^2 |g - g'|^{n-2} dV_{i,2} \\
&\leqslant [C_2' C_4' d_{\infty, t}(g, g')]^{n-2} \sum_i \int_0^t |g - g'|^2 dV_{i,2} \\
&\leqslant [C_2' C_4' d_{\infty, t}(g, g')]^{n-2} \sum_i \int_0^t |g - g'|^2 d\Lambda_i \\
&= 2d_{2,t}^2(g, g')[C_2' C_4' d_{\infty, t}(g, g')]^{n-2} \leqslant 2\frac{n!}{2}d_{2,t}^2(g, g')[C_2' C_4' d_{\infty, t}(g, g')]^{n-2}.
\end{aligned}
$$

We are now ready to apply Theorem 11. The argument is classic, see for example, in [26]. By Lemma 3 and Lemma 4, it suffices to prove

$$\Pr_M\left[M(T|g_{\widehat{a}}) \geqslant \widehat{H}^2\left(\frac{\widehat{a} + a}{2}, a\right) - \Delta \text{ and } \widehat{H}\left(\frac{\widehat{a} + a}{2}, a\right) > \frac{z}{4}\right] \leqslant \exp\left[-O(z)\right].$$

Let $\overline{b} := \frac{a+b}{2}$ for $b \in \mathcal{A}$. The left hand side of the above inequality is bounded by

$$
\begin{aligned}
&\Pr_M\left[M(T|g_b) \geqslant \widehat{H}^2(\overline{b}, a) - \Delta \text{ and } \widehat{H}(\overline{b}, a) > \frac{z}{4} \text{ for some } b\right] \\
&\leqslant \sum_{j=1}^\infty \Pr_M\left[M(T|g_b) \geqslant \left(2^{j-1}\frac{z}{4}\right)^2 - \Delta \text{ and } \widehat{H}(\overline{b}, a) > 2^j\frac{z}{4} \text{ for some } b\right].
\end{aligned}
$$

Denote the $j$-th term on the right hand side as $P_j$. Note that $g_a = 0$ and $M(T|g_b) = M(T|g_b - g_a)$, and $\widehat{H}(\overline{b}, a) = d_{2,T}^2(g_b, g_a)$. So we can apply Theorem 11 on $P_j$. By setting $z = \Omega(\max\{D, \Delta\})$ and $z_j = O(2^j z)$, we have $P_j \leqslant \exp\left[-z_j\right]$ and thus $\sum_{j=1}^\infty P_j \leqslant \exp\left[-O(z)\right]$. $\qquad\square$

## B.5 Sample Complexity of MLE for Learning Time Varying Coverage Functions

To apply Theorem 15 in our case, we need: 1) to bound the dimension of our hypothesis class; 2) to transfer the Hellinger distance to $\ell_2$ error to get the final bound.

**Lemma 16.** *The $(d_2, d_\infty)$-covering dimension of $\mathcal{G}$ is at most the number of random features $W$.*

*Proof.* Note that $d_{2,t}$ and $d_{\infty,t}$ are both nondecreasing with respect to $t$. So it suffices to show the existence of a covering of size exponential in $W$ with respect to both $d_{2,T}$ and $d_{\infty,T}$. In the following, we only consider the time $T$, and write $d_{2,T}$ ($d_{\infty,T}$ respectively) as $d_2$ ($d_\infty$ respectively). Note that

$$d_2^2(g_{a'}, g_{a''}) = \widehat{H}^2 \left( \frac{a' + a}{2}, \frac{a'' + a}{2} \right) \quad \text{and} \quad d_\infty(g_{a'}, g_{a''}) = \max_{t,S} \left| \frac{a' + a}{2a} - \frac{a'' + a}{2a} \right| = \left| \frac{a' - a''}{2a} \right|.$$

Then, the covering dimension of $\mathcal{G}$ is just the $(d_2, d_\infty)$-covering dimension of $\mathcal{A}$ on which the distances are (overloading notations):

$$d_2^2(a', a'') := d_2^2(g_{a'}, g_{a''}), d_\infty(a', a'') := d_\infty(g_{a'}, g_{a''}).$$

Then we can apply the same argument as Lemma 15 in [8] to show the dimension is at most $W$. That is, define a mapping from $\boldsymbol{w}$ to $a_{\boldsymbol{w}}^K$, and show that the $\ell_\infty$ distance of the former is approximately the $d_2$ distance of the latter, and the $d_\infty$ distance is bounded by the $d_2$ distance (up to constant factors).

We will need to introduce the following definition and then prove a claim showing that the $\ell_\infty$ distance on $\boldsymbol{w}$ is approximately the $d_2$ distance on $a_{\boldsymbol{w}}^K$.

**Definition 17.** *Define* $\xi = \min_{\boldsymbol{w} \neq 0} \frac{\boldsymbol{w}^\top \boldsymbol{A} \boldsymbol{w}}{\boldsymbol{w}^\top \boldsymbol{w}}$*, where* $\boldsymbol{A} = \frac{1}{2T} \sum_{\mathcal{S}} \mathbb{P}(\mathcal{S}) \boldsymbol{\Phi} \boldsymbol{\Phi}^\top$ *and*

$$\boldsymbol{\Phi} = \int_0^T \phi dt, \quad \text{and} \quad \phi = [K(t - t(\mathcal{S}, \boldsymbol{\tau}_1)), \dots, K(t - t(\mathcal{S}, \boldsymbol{\tau}_W)]^\top.$$

**Claim 8.** *For an* $\boldsymbol{w}, \boldsymbol{w}'$,

$$\sqrt{\frac{\xi}{2T a_{\max}^w}} \|\boldsymbol{w} - \boldsymbol{w}'\|_\infty \leqslant d_2(a_{\boldsymbol{w}}, a_{\boldsymbol{w}'}) \leqslant \frac{W \kappa_{\max}}{2\sqrt{a_{\min}^w}} \|\boldsymbol{w} - \boldsymbol{w}'\|_\infty.$$

*Proof.* (1) By definition, we have

$$d_2^2(a_{\boldsymbol{w}}, a_{\boldsymbol{w}'}) = \frac{1}{2} \mathbb{E}_{\mathcal{S}} \mathbb{E}_t \left[ \sqrt{\boldsymbol{w}^\top \phi} - \sqrt{\boldsymbol{w}'^\top \phi} \right]^2 = \frac{1}{2} \mathbb{E}_{\mathcal{S}} \mathbb{E}_t \left[ \frac{\boldsymbol{w}^\top \phi - \boldsymbol{w}'^\top \phi}{\sqrt{\boldsymbol{w}^\top \phi} + \sqrt{\boldsymbol{w}'^\top \phi}} \right]^2$$

$$\geqslant \frac{1}{2a_{\max}^w} \mathbb{E}_{\mathcal{S}} \mathbb{E}_t \left[ \boldsymbol{w}^\top \phi - \boldsymbol{w}'^\top \phi \right]^2$$

$$= \frac{1}{2a_{\max}^w T} (\boldsymbol{w} - \boldsymbol{w}')^\top \boldsymbol{A} (\boldsymbol{w} - \boldsymbol{w}')$$

$$\geqslant \frac{\xi}{2a_{\max}^w T} \|\boldsymbol{w} - \boldsymbol{w}'\|_2^2 \geqslant \frac{\xi}{2a_{\max}^w T} \|\boldsymbol{w} - \boldsymbol{w}'\|_\infty^2.$$

(2) By definition we have

$$d_2^2(a_{\boldsymbol{w}}, a_{\boldsymbol{w}'}) = \frac{1}{2} \mathbb{E}_{\mathcal{S}} \mathbb{E}_t \left[ \sqrt{\boldsymbol{w}^\top \phi} - \sqrt{\boldsymbol{w}'^\top \phi} \right]^2 = \frac{1}{2} \mathbb{E}_{\mathcal{S}} \mathbb{E}_t \left[ \frac{\boldsymbol{w}^\top \phi - \boldsymbol{w}'^\top \phi}{\sqrt{\boldsymbol{w}^\top \phi} + \sqrt{\boldsymbol{w}'^\top \phi}} \right]^2$$

$$\leqslant \frac{1}{4a_{\min}^w} \mathbb{E}_{\mathcal{S}} \mathbb{E}_t \left[ \boldsymbol{w}^\top \phi - \boldsymbol{w}'^\top \phi \right]^2$$

$$\leqslant \frac{1}{4a_{\min}^w} W^2 \kappa_{\max}^2 \|\boldsymbol{w} - \boldsymbol{w}'\|_\infty^2.$$

$\square$

To bound the dimension, the key is to construct coverings of small sizes. By the above claim, the $d_2$ metric on $\mathcal{A}$ approximately corresponds to the $\ell_\infty$ metric on the set of weights. So based on coverings for the weights with respect to the $\ell_\infty$ metric, we can construct coverings for $\mathcal{A}$ with respect to the $d_2$ metric. We then show that they are also coverings with respect to the $d_\infty$ metric. The bound on the dimension then follows from the sizes of these coverings.

More precisely, given $\epsilon > 0$ and a ball $\mathcal{B} \subseteq \mathcal{A}$ with radius $R \geqslant \epsilon$, we construct an $\epsilon$-covering $\mathcal{C}$ as follows. Define a mapping $\pi : \boldsymbol{w} \mapsto a_{\boldsymbol{w}}$, and define $\mathcal{B}^w = \pi^{-1}(\mathcal{B})$. By Claim 8, the radius of $\mathcal{B}^w$

is at most $R^w = \sqrt{\frac{2Ta^w_{\max}}{\xi}} R$ (with respect to the $\ell_\infty$ metric). Now consider finding an $\epsilon^w$-covering for $\mathcal{B}^w$ with respect to the $\ell_\infty$ metric, where $\epsilon^w = \left(\frac{W\kappa_{\max}}{2\sqrt{a^w_{\min}}}\right)^{-1}\epsilon$. Since $\mathcal{B}^w \subseteq \mathbb{R}^W$, by taking the grid with length $\epsilon^w/2$ on each dimension, we can get such a covering $\mathcal{C}^w$ with

$$|\mathcal{C}^w| \leqslant \left(\frac{4R^w}{\epsilon^w}\right)^W \leqslant \left(4\sqrt{\frac{2Ta^w_{\max}}{\xi}}\frac{W\kappa_{\max}}{2\sqrt{a^w_{\min}}}\frac{R}{\epsilon}\right)^W.$$

Let $\mathcal{C} = \pi(\mathcal{C}^w)$, and for any $b \in \mathcal{B}$ find $\widetilde{b}$ as follows. Suppose $\boldsymbol{w}_b \in \mathcal{B}^w$ satisfies $\pi(\boldsymbol{w}_b) = b$ and $\boldsymbol{w}_{\widetilde{b}}$ is the nearest neighbor of $\boldsymbol{w}_b$ in $\mathcal{C}^w$, then we set $\widetilde{b} = \pi(\boldsymbol{w}_{\widetilde{b}})$.

First, we argue that $\mathcal{C}$ is an $\epsilon$-covering w.r.t. the $d_2$ metric, *i.e.*, $d(b, \widetilde{b}) < \epsilon$ for any $b \in \mathcal{B}$. It follows from Claim 8:

$$d_2(b, \widetilde{b}) \leqslant \frac{W\kappa_{\max}}{2\sqrt{a^w_{\min}}}\|\boldsymbol{w}_b - \boldsymbol{w}_{\widetilde{b}}\|_\infty < \frac{W\kappa_{\max}}{2\sqrt{a^w_{\min}}}\epsilon^w = \epsilon.$$

Second, we argue that $\mathcal{C}$ is also an $O(\epsilon)$-covering w.r.t. the $d_\infty$ metric, *i.e.*, $d_\infty(b, \widetilde{b}) = O(\epsilon)$ for any $b \in \mathcal{B}$.

$$
\begin{aligned}
d_\infty(\pi(\boldsymbol{w}_b), \pi(\boldsymbol{w}_{\widetilde{b}})) &= \max_{t,\mathcal{S}}\left|\sqrt{\frac{b+a}{2a}} - \sqrt{\frac{\widetilde{b}+a}{2a}}\right| \\
&= \max_{t,\mathcal{S}}\left|\frac{|b-\widetilde{b}|}{\sqrt{2a}\left(\sqrt{b+a}+\sqrt{\widetilde{b}+a}\right)}\right| \\
&\leqslant \frac{\max_{t,\mathcal{S}}\left|(\boldsymbol{w}_{\widetilde{b}}-\boldsymbol{w}_{\widetilde{b}})^\top\phi\right|}{2\sqrt{2a_{\min}(a^w_{\min}+a_{\min})}} \\
&\leqslant \frac{W\kappa_{\max}}{2\sqrt{2a_{\min}(a^w_{\min}+a_{\min})}}\|\boldsymbol{w}_b - \boldsymbol{w}_{\widetilde{b}}\|_\infty.
\end{aligned}
$$

So the conditions in the definition of the dimension are satisfied with $D = W$, $c_1 = 4\sqrt{\frac{2Ta^w_{\max}}{\xi}}\frac{W\kappa_{\max}}{2\sqrt{a^w_{\min}}}$ and $c_2 = \frac{W\kappa_{\max}}{2\sqrt{2a_{\min}(a^w_{\min}+a_{\min})}}$, and thus the dimension of $\mathcal{A}$ is at most $W$. $\qquad\square$

Now, we can plug the lemma into Theorem 15, and convert the Hellinger distance to the $\ell_2$ distance between $f$ and our output function $\widehat{f}$ defined by $\widehat{a}$.

**Theorem 18.** *Suppose $\widehat{a}$ is an $\epsilon_\ell$-MLE, and $\widehat{f}$ is the corresponding function.*
*(i) Suppose there exists an $\widetilde{a} \in \mathcal{A}$ such that with probability at least $1-\delta_1$ over $\mathcal{S}$, $\mathbb{E}_t\,(a'-a)^2 \leqslant \epsilon_a^2$.*
*Then for any $0 \leqslant t \leqslant T$, and $\nu > 0$,*

$$
\begin{aligned}
&\mathbb{E}_\mathcal{S}\left[\widehat{f}(\mathcal{S},t)-f(\mathcal{S},t)\right]^2 \\
&\leqslant O\left(t^2\left\{\nu A_{\max}^2 + \frac{A_{\max}}{mT}\left[W + \log\frac{1}{\nu} + \epsilon_\ell\right] + A_{\max}\left[\epsilon_a + \frac{\epsilon_a^2}{A_{\min}}\log\left(\frac{a_{\max}}{a^w_{\min}}\right) + \sqrt{\frac{c_\ell^2\epsilon_a^2}{A_{\min}mT}\log\frac{1}{\nu}}\right]\right\}\right)
\end{aligned}
$$

*where $A_{\max} = a_{\max} + a^w_{\max}$, $A_{\min} = a_{\min} + a^w_{\min}$, and $c_\ell^2$ is defined in Lemma 13.*
*(ii) Consequently, if*

$$W = O\left((CZ\kappa_{\max})^2\left[\left(\frac{A_{\max}T}{\epsilon}\right)^{5/2} + \left(\frac{A_{\max}T\log\frac{a_{\max}}{a^w_{\min}}}{\epsilon A_{\min}}\right)^{5/4}\right]\log\frac{mA_{\max}T}{\epsilon\delta}\right)$$

*and*

$$m = O\left(\frac{A_{\max}T}{\epsilon}\left[W + \log\frac{A_{\max}T}{\epsilon} + \epsilon_\ell\right] + \frac{1}{A_{\min}\sqrt{a^w_{\min}}T}\log\frac{A_{\max}T}{\epsilon}\right).$$

*then with probability* $\geqslant 1 - \delta$ *over* $\{\tau_i\}_{i=1}^{W}$, *for any* $0 \leqslant t \leqslant T$,

$$\mathbb{E}_{\mathcal{S}} \left[ \widehat{f}(\mathcal{S}, t) - f(\mathcal{S}, t) \right]^2 \leqslant \epsilon.$$

*Proof.* (i) By Theorem 15 and Lemma 13, there exists $\Omega_{\mathcal{S}}$ of probability at least $1 - m\delta_1$ so that for any outcome of $\{\mathcal{S}_i\}_{i=1}^{m}$ in it, we have that with probability $\geqslant 1 - 2\delta_2$,

$$\widehat{H}^2(\widehat{a}, a) \leqslant z = O\left( D + B(\delta_2) + \epsilon_\ell + \log \frac{1}{\delta_2} \right)$$

where $D \leqslant W$ by Lemma 16.

Since $\widehat{H}^2(\widehat{a}, a) \leqslant mT(a_{\max} + a_{\max}^w)$ and $\mathbb{E}_{\mathcal{D}^m}\left[ \widehat{H}^2(\widehat{a}, a) \right] = mTh^2(\widehat{a}, a)$, we have

$$h^2(\widehat{a}, a) \leqslant \epsilon^2(\delta_1, \delta_2) := (1 - m\delta_1)(1 - 2\delta_2)\frac{z}{mT} + (m\delta_1 + 2\delta_2)(a_{\max} + a_{\max}^w).$$

Now we convert the Hellinger distance between the intensities to the $\ell_2$ distance between the funciton $f$ and the output $\widehat{f}$ defined by $\widehat{a}$. For any $0 \leqslant \tau \leqslant T$,

$$
\begin{aligned}
\mathbb{E}_{\mathcal{S}} \left[ \widehat{f}(\mathcal{S}, \tau) - f(\mathcal{S}, \tau) \right]^2 &\leqslant \mathbb{E}_{\mathcal{S}} \left[ \int_0^\tau |\widehat{a}(\mathcal{S}, t) - a(\mathcal{S}, t)| \, dt \right]^2 \\
&\leqslant \tau \mathbb{E}_{\mathcal{S}} \int_0^\tau [\widehat{a}(\mathcal{S}, t) - a(\mathcal{S}, t)]^2 \, dt \\
&\leqslant \tau \mathbb{E}_{\mathcal{S}} \int_0^\tau \left[ \left( \sqrt{\widehat{a}(\mathcal{S}, t)} - \sqrt{a(\mathcal{S}, t)} \right) \left( \sqrt{\widehat{a}(\mathcal{S}, t)} + \sqrt{a(\mathcal{S}, t)} \right) \right]^2 dt \\
&\leqslant 2(a_{\max} + a_{\max}^w)\tau \mathbb{E}_{\mathcal{S}} \int_0^\tau \left[ \sqrt{\widehat{a}(\mathcal{S}, t)} - \sqrt{a(\mathcal{S}, t)} \right]^2 dt \\
&\leqslant 4(a_{\max} + a_{\max}^w)\tau^2 h^2(\widehat{a}, a) \leqslant 4(a_{\max} + a_{\max}^w)\tau^2 \epsilon^2(\delta_1, \delta_2).
\end{aligned}
$$

The first statement then follows from choosing $\delta_1 = \nu/m$ and $\delta_2 = \nu$.

(ii) The second statement follows from the first statement and Lemma 2. More precisely, we check each error term and set the parameters as follows.

- To ensure $t^2 A_{\max}^2 \nu = O(\epsilon)$, let $\nu = O\left( \frac{\epsilon}{A_{\max}^2 T^2} \right)$.

- To ensure $\frac{t^2 A_{\max}}{mT} \left[ W + \log \frac{1}{\nu} + \epsilon_\ell \right] = O(\epsilon)$, let

$$m = O\left( \frac{A_{\max}T}{\epsilon} \left[ W + \log \frac{A_{\max}T}{\epsilon} + \epsilon_\ell \right] \right). \tag{17}$$

- To ensure that $\epsilon_a^2 = O(\epsilon_0^2)$, let $\sigma = \sqrt{\epsilon_0}$, and

$$W = O\left( \left( \frac{CZ\kappa_{\max}}{\epsilon_0 \sigma} \right)^2 \log \frac{1}{\delta_1 \delta} \right) = O\left( \frac{(CZ\kappa_{\max})^2}{\epsilon_0^{5/2}} \log \frac{mA_{\max}T}{\epsilon \delta} \right).$$

- To ensure $t^2 A_{\max}\epsilon_a = O(\epsilon)$, we need $\epsilon_0^2 = O\left( \left( \frac{\epsilon}{A_{\max}T} \right)^2 \right)$. To ensure $\frac{t^2 A_{\max}\epsilon_a^2}{A_{\min}} \log \left( \frac{a_{\max}}{a_{\min}^w} \right) = O(\epsilon)$, we need $\epsilon_0^2 = O\left( [A_{\min}\epsilon] \bigg/ \left[ A_{\max}T^2 \log \left( \frac{a_{\max}}{a_{\min}^w} \right) \right] \right)$. Then we need

$$W = O\left( (CZ\kappa_{\max})^2 \left[ \left( \frac{A_{\max}T}{\epsilon} \right)^{5/2} + \left( \frac{A_{\max}T \log \frac{a_{\max}}{a_{\min}^w}}{\epsilon A_{\min}} \right)^{5/4} \right] \log \frac{mA_{\max}T}{\epsilon \delta} \right). \tag{18}$$

- To ensure $t^2 A_{\max} \sqrt{\frac{c_\ell^2 \epsilon_a^2}{A_{\min} m T}} \log \frac{1}{\nu} = O(\epsilon)$, we need

$$m = O\left(\frac{c_\ell^2}{A_{\min} T} \log \frac{A_{\max} T}{\epsilon}\right) = O\left(\frac{1}{A_{\min}\sqrt{a_{\min}^w} T} \log \frac{A_{\max} T}{\epsilon}\right). \qquad (19)$$

The bound for $W$ and $m$ then follows from (18) and (17) (19) respectively. The kernel bandwidth $\sigma$ is chosen such that $\sigma = \sqrt{\epsilon_0} = O\left(\min\left\{\left(\frac{\epsilon}{A_{\max} T}\right)^{1/2}, [A_{\min}\epsilon]^{1/4} \Big/ \left[A_{\max} T^2 \log\left(\frac{a_{\max}}{a_{\min}^w}\right)\right]^{1/4}\right\}\right).$

$\square$