[Reviews · NeurIPS 2014]

Submitted by Assigned_Reviewer_9

This paper defines temporal coverage functions, a generalization of coverage functions with dependence. It introduces a method to learn them from previous time history. It also includes experimental validation on real data sets.

The paper is well written, and clear. It is also very technical, and was not an easy read for me. Judging its quality is even harder. I think it would help if the authors could demonstrate with an example what a time-varying coverage function means in practice, for a given data set. That would help me refine the quality score I given in my review.

It is not clear to me which of the baseline algorithms in section 7.1 were also using time information, so it would be great it the authors could clarify this? I believe that for a fair comparison the baseline algorithm should use time information as well.

Summary: This paper defines temporal coverage functions, a generalization of coverage functions with dependence. It introduces a method to learn them from previous time history. It also includes experimental validation on real data sets.

Submitted by Assigned_Reviewer_29

This paper studies the task of learning time-varying coverage functions by modeling the cumulative intensity function of a counting process where time-varying coverage function controls the propensity of occurring events over time. They parametrize the intensity function as a weighted combination of random kernel functions and learn the parameters by maximum likelihood estimation. Their algorithm is guaranteed to learn the the time-varying coverage function only from polynomial number of training examples. Finally, they validate the proposed model on influence estimation and maximization problems with both synthetic and real-world cascade data from information diffusion.

Summary: A well-written, theoretically sound paper that studies an important learning problem in submodular coverage. Their model is supported by detailed proofs and experiments.

Submitted by Assigned_Reviewer_42

First of all, I have to apologize to the authors that this paper has been reviewed by non-experts like me.

This paper deals with a problem of estimating coverage functions dependent of the temporal domain, and proposes a (probably) novel method for estimating such a time-dependent coverage function. The formulation based on a counting process looks technically sound and well motivated.

Basically, this paper is well written and clear, and its organization is no problem. Since the main contribution of this paper lies on a well formulated algorithm with some theoretical minds, it might be OK that connections between the proposed method and some practical applications seems to be a bit weak.

I would like to know some relationships to some other counting processes such as Hawkes processes.
Summary: This paper deals with a problem of estimating coverage functions dependent of the temporal domain, and proposes a (probably) novel method for this purpose. The formulation based on a counting process looks technically sound and well motivated.
Author Feedback
Author rebuttal: We thank the reviewers for valuable and timely comments.

First, we'd like to emphasize the challenges and contributions:

1. Learning coverage functions alone is already very challenging in that there are an exponential number of sets to consider, and both the ground set and the weights are unknown. Many existing algorithms are mostly of theoretical interest and impractical for real world problems. Our paper addresses an even harder problem of learning such combinatorial functions with temporal structures, which is new and has not been addressed before. Such functions have important applications in social network analysis and algorithmic game theory.

2. We develop a novel parameterization of the time-varying coverage function by illustrating its key connections with counting processes for the first time.

3. We present an efficient algorithm which provably learns the parameters by maximum likelihood estimation, and provide rigorous theoretical analysis of its sample complexity.

4. Experiments on both synthetic and real datasets show that our method can achieve significantly better performance over existing approaches when the latent true diffusion model is misspecified or unknown.

Second, we'd like to explain more on real world applications (in response to Assigned_Reviewer_42 and Assigned_Reviewer_9):

1. Time-varying coverage functions arise in many interesting applications:

In information diffusion, they model the influence of a set of users in a social network over time. With this function, marketers can decide which set of influential users to choose in order to achieve the goal of an ads campaign before some due date.

In algorithmic game theory, they represent the valuations of a bundle of items to user. The higher the valuation, the more likely a user will take frequent actions. For instance, a particular discussion forum is configured to a particular set of settings. Depending on the configuration, it may engage users better and trigger more user actions (clicking, browsing, and reply to posts). The platform host would like to use this learned function to guide the default choice of setting.

In facility location management, they represent the utilities of a group of locations to open bank accounts or build warehouses, which is expected to save costs over time. As a consequence, learning of the time-varying coverage functions allows us to query the value of a set S of items along time to strategically maximize our target utilities.

2. We will elaborate the application to information diffusion below:

In this application, we seek to find a set of earlier adopters to give promotions such that they can trigger the largest expected number of follow-ups by time t in the future.

A key step in solving this problem is to calculate the influence value (the expected number of infected nodes) at time t of a given set S of source nodes, which is a time-varying coverage function dependent on S and t.

In recent literature, a commonly used paradigm is to first fit a diffusion model up to a given time t, and then calculate the influence based on the learned model. In this two-stage paradigm, the calculated influence is determined by the underlying assumed diffusion model. However, in reality, the true diffusion mechanism is never known, and the proposed model may be misspecified.

In contrast, the direct learning of the time-varying coverage function allows us to query the influence value of any given source set S at any time t without assuming any specific diffusion model and the need of updating and refitting.

Assigned_Reviewer_9:

We model the intensity function of the counting process as a weighted combination of random kernel functions. The kernels are centered around randomly chosen time points, rather than on previous event time points. Thus, the kernels are not used to capture the self-exciting effect (as in Hawkes process), but to obtain a smooth intensity function.

Assigned_Reviewer_42:

All competitors in section 7.1 use the temporal information. For the kernel regression, it is fitted to each specific time point at which we compare the estimated influence value. For the two-stage approaches, a diffusion model is first fitted up to time t and then the influence value is calculated based on the model. In fact, the two-stage approaches require even more information, such as the identity of the infected node, to learn the model. This information is not needed for our method, but we give the competitors this advantage in the experiments.